# Observed impacts of aerosol concentration on maritime tropical convection within constrained environments using airborne radiometer, radar, lidar, and dropsondes

Corey G. Amiot[1,#], Timothy J. Lang[2], Susan C. van den Heever[3], Richard A. Ferrare[4], Ousmane O. Sy[5], Lawrence D. Carey[1], Sundar A. Christopher[1], John R. Mecikalski[1], Sean W. Freeman[3,##], George Alexander Sokolowsky[3,###], Chris A. Hostetler[4], and Simone Tanelli[5]

[1]Department of Atmospheric and Earth Science, The University of Alabama in Huntsville, Huntsville, AL, 35899, USA
[2]NASA Marshall Space Flight Center, Huntsville, AL, 35812, USA
[3]Department of Atmospheric Science, Colorado State University, Fort Collins, CO, 80523, USA
[4]NASA Langley Research Center, Hampton, VA, 23681, USA
[5]Jet Propulsion Laboratory, California Institute of Technology, Pasadena, CA, 91109, USA

[#]Now at NASA Postdoctoral Program, NASA Marshall Space Flight Center, Huntsville, AL, 35812, USA
[##]Now at Department of Atmospheric and Earth Science, The University of Alabama in Huntsville, Huntsville, AL, 35899, USA
[###]Now at Verisk Analytics, Inc., Boston, MA, 02111, USA

*Correspondence to*: Corey G. Amiot (corey.g.amiot@nasa.gov)

**Abstract.** Aerosol modulation of atmospheric convection remains an important topic in ongoing research. A key challenge in evaluating aerosol impacts on cumulus convection is isolating their effects from environmental influences. This work investigates aerosol effects on maritime tropical convection using airborne observations from NASA's Cloud, Aerosol and Monsoon Processes Philippines Experiment (CAMP$^2$Ex). Eight environmental parameters with known physical connections to cloud and storm formation were identified from dropsonde data, and 92 dropsondes were matched with corresponding CAMP$^2$Ex flight "scenes." To constrain environmental conditions, scenes were binned based on their association with "low," "medium," or "high" values for each dropsonde-derived parameter. In each scene and environmental bin, eight radar- and radiometer-based parameters directly related to convective intensity and/or prevalence were correlated with lidar-derived aerosol concentrations to examine trends in convective characteristics under different aerosol conditions. Threshold values used to stratify the environments were varied across four sensitivity tests to examine how the convective-aerosol correlations within each environmental bin responded. The results were generally inconclusive, with relatively weak correlations observed with limited statistical significance in many cases. Some interesting and potentially impactful comparisons identified in the convective-aerosol analyses support the idea of warm-phase convective invigoration trends and suggest that higher aerosol concentrations were correlated with stronger and/or more-prevalent convection in some cases, while other cases saw a "Goldilocks" zone of medium aerosol concentration favoring enhanced convection. Our results also stress the importance of considering environmental conditions when evaluating aerosol impacts.

**Short summary**

Decoupling aerosol and environmental impacts on convection is challenging. Using airborne data, we correlated convective metrics with aerosol concentrations in several different environments. Results were mixed, but some comparisons suggest that medium-to-high aerosol concentrations were occasionally strongly correlated with convective intensity and prevalence, especially when the atmosphere was relatively unstable. Storm environment is important to consider when evaluating aerosol effects.

## 1. Purpose and background

The primary purpose of this study is to explore potential impacts of aerosol concentration on maritime tropical convection during NASA's Cloud, Aerosol and Monsoon Processes Philippines Experiment (CAMP²Ex) from a remote-sensing perspective while considering adjacent environmental conditions. Aerosol influences on convection have been a significant research topic and, as detailed further in section 2, were one of the main science foci of CAMP²Ex. These aerosol effects impact other meteorological phenomena and have broader consequences for society, and therefore warrant considerable research attention. However, as we describe throughout this section, isolating aerosol impacts on convection from those of other (e.g., environmental) effects is challenging and requires additional study despite the wide range of past works that have investigated this topic in detail.

Increased aerosol concentration is generally associated with increased cloud condensation nuclei, with aerosol size distribution influencing cloud particle size distribution (Junge and McLaren, 1971). In shallow clouds, the second indirect effect of aerosols describes a decrease in precipitation formation and increase in cloud lifetime (Albrecht, 1989), resulting from reduced cloud droplet sizes due to increased competition for water vapor (e.g., Rosenfeld and Lensky, 1998; Sherwood, 2002). However, precipitation-sized hydrometeors that form in higher aerosol concentrations may be larger, owing to ample cloud droplets available for collection and droplet growth (e.g., Stroud et al., 2007; Altaratz et al., 2008; Saleeby et al., 2010). These changes in cloud coverage and precipitation reaching the surface can have a profound impact on the total radiative forcing in the atmosphere (e.g., Sokolowsky et al., 2022) and direct consequences on society (e.g., modulation of local rainfall patterns; Berg et al., 2008).

Many research efforts have explored aerosol invigoration of tropical convection (i.e., increases in updraft velocity), including the studies and formative discussions by Andreae et al., (2004) and Rosenfeld et al., (2008) regarding the positive and detrimental impacts of aerosols on convection. A key factor in evaluating these aerosol impacts is considering whether their strongest influences are realized in regions where condensational latent heat release is most profound below the environmental 0 °C level (i.e., warm-phase invigoration) or in regions where latent heat release via freezing occurs above the environmental 0 °C level (i.e., cold-phase invigoration) instantaneously during storm development (e.g., Igel and van den Heever, 2021). Warm-phase invigoration (Andreae et al., 2004) depends on the finite supersaturation within a cloud's updraft, and if this supersaturation is in a quasi-equilibrium state the rate of condensation onto available aerosols will depend only on the updraft's velocity (Grabowski and Morrison, 2020). As a result, if the concentration of cloud droplets increases, the quasi-equilibrium supersaturation decreases, which leads

to an increase in buoyancy, updraft velocity, and condensation rate within the cloud (Grabowski and Morrison 2020).

Cold-phase invigoration (Rosenfeld et al., 2008) depends on the off-loading of frozen precipitation aloft in the cloud, since the latent heat release from freezing will only balance the mass loading of any liquid hydrometeors lofted above the environmental 0 °C level, and any buoyancy increases owing to the formation and fallout of precipitation will likely be negligible (Grabowski and Morrison 2020).

The impacts of aerosol warm-phase and cold-phase invigoration of convection have received mixed results in prior observational and numerical modeling studies. Regarding warm-phase invigoration, for example, van den Heever et al., (2006) noted stronger updrafts associated with higher aerosol concentrations within a variety of aerosol profiles. Sheffield et al., (2015) explained how enhanced aerosol concentrations can increase cloud water content and produce more-vigorous updrafts via latent heat of condensation. Likewise, Marinescu et al., (2021) noted a 5–15% increase in mean updraft velocity around 4–7 km AGL when cloud condensation nuclei concentrations were relatively high. Observational studies have also explored the impacts of increased aerosol concentrations on convection. For example, Lin et al., (2006) found enhancements in cloud height and cloud cover associated with increased aerosol concentrations in the Amazon. Likewise, Fan et al., (2018) identified increases in convective strength owing to the activation of additional cloud condensation nuclei in regions with high concentrations of ultrafine aerosols in the Amazon. Using a combination of observations and simulations, Zhang et al., (2023) described how fine aerosols enhance convection which, in turn, modulates the surrounding environment and feeds back into larger-scale atmospheric circulations. Studies have also suggested that smaller cloud droplets associated with higher aerosol concentrations may enhance updraft/convective intensity via increased latent heat released during freezing and enhanced depositional growth above the environmental 0 °C level (e.g., van den Heever and Cotton, 2007; Rosenfeld et al., 2008) in association with cold-phase invigoration. However, others have demonstrated that convective intensity increases are primarily driven by low-level condensational heating, rather than freezing above the environmental 0 °C level (Igel and van den Heever, 2021; Cotton and Walko, 2021), further indicating the importance of evaluating aerosol concentrations within/around warm-phase regions.

In contrast, other studies (e.g., Grabowski and Morrison, 2016; Grabowski and Morrison, 2020; Varble, 2023) have presented evidence that increased aerosol concentrations do not invigorate convection above the environmental freezing level but may do so below the environmental freezing level. As mentioned previously, the former results from increased positive buoyancy, via enhanced latent heat of freezing, being offset by increased negative buoyancy, via mass loading from a greater liquid water content being lofted above the environmental freezing level (Grabowski and Morrison, 2020), while the latter results from higher aerosol concentrations leading to lower supersaturation values within the surrounding environment and increased buoyancy, which in turn lead to higher updraft velocities and enhanced latent heating associated with increased condensation (e.g., Grabowski and Morrison, 2017; Grabowski and Morrison, 2020). The results of Öktem et al., (2023) appear to contradict the study by Fan et al., (2018) by explaining how they identified no statistically significant relationship between aerosol concentration and convective intensity.

Therefore, the role of enhanced aerosol concentration on updraft velocity can strongly depend on whether they are within regions wherein warm-phase or cold-phase processes dominate.

Many studies have also identified situations where higher aerosol concentrations may be detrimental for convection. For instance, entrainment of relatively dry environmental air may cause rapid evaporation of smaller cloud droplets associated with higher aerosol concentrations, decreasing cloud/storm structure (e.g., Liu et al., 2016). Veals et al., (2022) noted a tendency for weaker convection in the presence of higher aerosol concentrations in central Argentina. This raises several questions about the true impact of increased aerosol concentrations on convection, which motivates our study herein. These differences in the outcomes of past studies also indicate that a "Goldilocks" zone of medium aerosol concentration may favor the strongest convection (e.g., Sokolowsky et al., 2022). Further, other works have discussed considerable difficulty in separating aerosol influences from atmospheric dynamics (e.g., Grabowski, 2018). Limitations in past numerical and observational studies are summarized by Varble et al., (2023), which highlights the continued uncertainty surrounding aerosol impacts on convection and further motivates our study.

Direct measurements of updraft vertical velocity allow for quantitative assessment of convective intensity. However, numerous remote sensing metrics serve as indirect indicators of convective intensity; these metrics and the implied microphysical processes in relation to convective intensity will be the primary focus of this study, as outlined further in the next paragraph. All else being equal, as an updraft fuels vertical development within a warm-phase cloud, continued condensation onto available cloud condensation nuclei will increase the cloud liquid water content and lead to the formation and growth of precipitation-size hydrometeors (e.g., via collision-coalescence). These increases in hydrometeor concentration and, especially, mean hydrometeor diameter within a given cloud particle size distribution can be identified by increases in equivalent radar reflectivity factor ($Z_H$) over the same region (Rinehart, 2010). Likewise, increased emission from these warm-phase hydrometeors will lead to increases in radiometer-retrieved brightness temperature ($T_b$) values (e.g., Spencer et al., 1994). If updraft velocity increases, these cloud and precipitation processes and their appearances within remote sensing products will likewise be enhanced (again, all else being equal). Additional hydrometeor formation and growth can also occur in the cold-phase region as vertical cloud development continues. As a result, increases in selected radar- and radiometer-derived products like $Z_H$ and $T_b$ can serve as indirect indicators of convective intensity.

Based on these studies, the primary science question we address is: How do radiometer- and radar-based metrics of storm intensity ("intensity" referring to peak updraft velocity) and prevalence vary with lidar-based observations of aerosol concentration when binned into similar environmental groups throughout CAMP²Ex? It should be noted that we evaluate these metrics as indicators of storm intensity, but updraft vertical velocity is not specifically calculated in this study. The results of these analyses are important as they provide insight into science questions for a major NASA field campaign, have relevance to upcoming NASA missions [e.g., Atmosphere Observing System (AOS, 2022)], and contribute knowledge to long-standing questions of aerosol influences on convection. We hypothesized that integrated cloud liquid water path (CLW), peak $Z_H$, peak Ku-/Ka-band radar dual-frequency ratio (DFR), and abundance of $Z_H$

observations $\geq$ 30 dBZ in a given scene would all increase under higher aerosol concentrations within an environmental group. These hypotheses were based on expectations that increased aerosol concentrations would favor development of smaller and more-numerous cloud droplets, invigorating convection and enhancing CLW, while the presence of fewer but larger raindrops would increase maximum $Z_H$ and overall presence of $Z_H \geq$ 30 dBZ along with greater Ka-band attenuation compared to Ku band (i.e., increased maximum DFR), further indicating enhanced convection. In addition, we hypothesized that radar- and radiometer-based metrics of storm intensity and prevalence would all increase within more-favorable environments, as revisited at the end of section 2, though this investigation is secondary to our aerosol analyses in this study. That is, given the environmental stratification methods employed as discussed in the next section, our focus was primarily on the correlations between convective and aerosol metrics and secondarily on the convective patterns associated with environmental variations. However, in each of these analyses, it is essential to note that correlation does not necessarily indicate causality, as a correlation between two variables may exist entirely due to indirect effects (e.g., Lin et al., 2006). In addition, it must be acknowledged that these radar- and radiometer-based metrics of convective intensity may vary due to factors not specifically owing to changes in peak updraft intensity (e.g., cloud microphysics; Varble et al., 2023). Despite these and other inherent difficulties, limitations, and uncertainties associated with separating aerosol and environmental influences on convection (e.g., Grabowski, 2018; Varble, 2018), the results herein provide important insight regarding observations of aerosol influences on the radar- and radiometer-based indicators of convective intensity. Additionally, potential trends found in the CAMP[2]Ex dataset could provide useful information to support future work. Section 2 covers the data and methods used, with Sects. 3 and 4 highlighting environmental stratification and aerosol analyses from the microwave-frequency datasets. Section 5 presents a summary, discussion of limitations, and future work.

## 2. Data and analysis methods

This section opens with a brief overview of the CAMP[2]Ex field campaign followed by a description of the datasets used and the uncertainties associated with those data. Discussions are then provided of the methods employed in the analyses of the environmental, convective, and aerosol parameters examined in this study.

### 2.1 CAMP[2]Ex

The field phase of CAMP[2]Ex occurred during 25 August – 10 October 2019, with the NASA P-3B Orion (P-3) aircraft conducting 19 science flights and the Stratton Park Engineering Company Learjet aircraft conducting 13 science flights out of Clark International Airport in the Philippines (Reid et al., 2023). The primary goal of CAMP[2]Ex was to evaluate the role of aerosols within Southeast Asia's monsoon systems by simultaneously examining aerosol characteristics alongside cloud and radiation properties (Reid et al., 2023). Out of the 19 P-3 science flights, 12 were associated with southwest monsoon conditions and seven were associated with the northeast monsoon within the Maritime Continent (Reid et al., 2023). A broad range of environmental, radiation, and aerosol conditions were observed throughout CAMP[2]Ex, along with numerous cloud types (i.e., shallow cumulus, congestus, and altostratus clouds) and convective organization (e.g., isolated convection and squall lines) (Reid et al., 2023). Data collected during CAMP[2]Ex included in situ and remote sensing datasets from the two aforementioned aircraft along with

ground-based and ship-based observation platforms, in addition to several numerical models (Reid et al., 2023). As will be discussed in the next subsection, our study primarily uses in situ and remote sensing data from the P-3 aircraft and falls under the CAMP²Ex science question of "To what extent are aerosol particles responsible for modulating warm and mixed-phase precipitation in tropical environments?", while also having direct implications for impacts on deeper convection and cloud meteorology (ESPO, 2020; Reid et al., 2023).

## 2.2 Datasets and their uncertainties

Key instruments for this study that were flown on the P-3 during CAMP²Ex include: the Advanced Microwave Precipitation Radiometer (AMPR; Spencer et al., 1994; Amiot et al., 2021), Airborne Precipitation and cloud Radar 3rd Generation (APR-3; Durden et al., 2020b), High Spectral Resolution Lidar 2 (HSRL2; Burton et al., 2016), and Advanced Vertical Atmospheric Profiling System (AVAPS; Hock and Young, 2017). All AMPR, APR-3, AVAPS, and HSRL2 data were gathered from the CAMP²Ex data repository (Aknan and Chen, 2020). Due to the direct correlations between cloud condensation nuclei concentration and lidar extinction (Ext), backscatter (Bsc), and aerosol optical thickness (AOT), all three parameters were analyzed from HSRL2's 355- and 532-nm channels that employ the HSRL technique (Hair et al., 2008), though 532-nm backscatter was of particular interest based on discussions in Lenhardt et al., (2022). The same quality control processes outlined in Amiot (2023) for the AMPR, APR-3, and AVAPS data were applied for this study, including application of AMPR's multiple data quality flags. As detailed in Lang et al., (2021), AMPR data were masked if: P-3 pitch or roll magnitude was $\geq 2°$, AMPR was in its nadir-stare mode, P-3 GPS altitude was < 3 km, the given AMPR scan included at least one pixel over land, and/or precipitation was present based on AMPR $T_b$ thresholding. Further, as discussed in Amiot (2023), 10 APR-3 files were removed due to high levels of noise, three dropsondes were excluded due to lack of corresponding AMPR data, and seven dropsondes were removed during data quality control (e.g., due to lack of recorded near-surface winds). Additionally, starting with the initial 144 dropsondes examined in Amiot (2023), a test was performed in this study to determine whether each dropsonde passed through cloud. Given the 3% uncertainty in AVAPS relative humidity (Freeman et al., 2020), any dropsonde where more than 20% of the dropsonde profile was associated with relative humidity > 97% was removed from the analysis, which amounted to five dropsondes in total. The HSRL2 data were screened for clouds (Hostetler, 2020) to avoid potential contamination of the aerosol analyses (e.g., Liu et al., 2016). Uncertainty values associated with each instrument were deemed negligible for this study. More specifically, AMPR's CLW root-mean-square deviation and median absolute deviation are both on the order of $10^{-2}$ kg m$^{-2}$ (Amiot, 2023) and AMPR's noise-equivalent differential temperature is 0.5–1.0 K (Amiot et al., 2021). APR-3's Ku-band (Ka-band) calibration uncertainty is roughly 1 dB (1.5 dB) (Durden et al., 2020b). The uncertainties in AVAPS's temperature, relative humidity, and pressure measurements are 0.2 °C, 3%, and 0.5 hPa, respectively (Freeman et al., 2020). Under typical conditions, the total systematic error for the HSRL2 532-nm extinction coefficient is estimated to be < 0.01 km$^{-1}$, which is within the typical state-of-the-art systematic error at visible wavelengths (Schmid et al., 2006; Rogers et al., 2009). The overall systematic error for HSRL2 backscatter calibration is estimated to be < 3% and the random errors for all aerosol products are typically < 10% for the backscatter and depolarization ratios (Hair et al., 2008; Ferrare et al., 2023).

## 2.3 Environmental, convective, and aerosol parameters

Eight environmental parameters with known physical connections to convective intensity were subjectively chosen for this study based on their ability to be fully captured by a statistically significant number of CAMP[2]Ex dropsondes. The eight selected parameters were: *modified normalized* (described below) Convective Available Potential Energy (CAPE); Lifting Condensation Level (LCL) altitude; K-Index; 850–700-, 850–500-, and 700–500-hPa temperature lapse rate (hereafter simply "lapse rate", LR); mean dew point temperature ($T_d$) below the 925-hPa level; and mean $T_d$ below 1 km AGL, which are hereafter referred to by their symbols in Table 1 and discussed in more detail below. A significant challenge in evaluating aerosol impacts on convection is to isolate aerosol influences from other sources of convection modulation, such as atmospheric dynamics, thermodynamics, and cloud microphysical processes (e.g., Liu et al., 2016; Grabowski 2018). Since a given convective plume will be affected by synoptic-scale (> 2000 km), mesoscale (2–2000 km), and sub-mesoscale (< 2 km) dynamics (Orlanski, 1975) and environmental conditions, it is important to understand and constrain environmental conditions associated with any convective element (herein "storm") of interest. Several environmental factors with direct physical connections to convection can be evaluated from remote-sensing and in situ observation platforms. Studies have demonstrated the utility of radiosonde data, the principles of which can be applied to dropsondes (e.g., AVAPS) to the extent offered by the dropsonde's launch altitude. CAPE is a measure of parcel buoyancy that can be used to diagnose potential updraft velocity. However, since CAPE is related to integrated buoyancy between the level of free convection and equilibrium level, an issue arises with computing CAPE from AVAPS during CAMP[2]Ex; since the P-3 did not fly above the equilibrium level during any science flight, the dropsondes did not capture the full vertical buoyancy profile associated with traditional CAPE. As such, the term "modified CAPE" is used herein and is defined mathematically as

$$\text{CAPE}_{\text{mod}}\left(\text{J kg}^{-1}\right) = g \int_{z_{lfc}}^{z_{P3}} \frac{(T_v - T_{v,0})}{T_{v,0}}\, dz, \tag{1}$$

where $g$ is gravitational acceleration; $T_v$ and $T_{v,0}$ are parcel and environmental virtual temperatures, respectively; $z$ is altitude; $z_{lfc}$ is the level of free convection; and $z_{P3}$ is P-3 altitude (Markowski and Richardson, 2010). With this definition, modified CAPE would be less than true CAPE within the same environment, which limits evaluation of parcel buoyancy. Since the dropsondes were often launched when the P-3 altitude was > 4 km AGL (Vömel et al., 2020), the instability indicated by modified CAPE can be compared across the environments. Despite this, P-3 altitude would have a direct effect on modified CAPE calculated via Eq. (1), with lower altitude (e.g., around 4 km AGL) biased toward lower modified CAPE by virtue of the dropsonde capturing a lesser vertical extent of the parcel buoyancy. To mitigate this effect, we normalized the CAPE via dividing by the dropsonde launch altitude, which yields (modified) normalized CAPE (Blanchard, 1998) via the relation

$$\text{NCAPE}_{\text{mod}}\left(\text{m s}^{-2}\right) = \frac{\text{CAPE}_{\text{mod}}}{z}, \tag{2}$$

where z is dropsonde launch altitude. In addition to normalizing the CAPE profiles by dropsonde launch altitude, an added benefit of examining $\text{NCAPE}_{\text{mod}}$ is that its units are m s$^{-2}$ as shown in Eq. (2), allowing direct evaluation of vertical acceleration over the dropsonde layer (Blanchard, 1998).

**Table 1: List of symbols used to represent the environmental, convective, and aerosol variables examined in this study, along with their units and a brief description of each variable.**

| Symbol | Units | Type | Description |
|:---:|:---:|:---:|:---:|
| $NCAPE_{mod}$ | m s$^{-2}$ | Environmental | Modified Normalized Convective Available Potential Energy |
| LCL | m | Environmental | Lifting Condensation Level altitude |
| K-Index | °C | Environmental | K-Index value |
| $LR_{850-700}$ | °C km$^{-1}$ | Environmental | Temperature lapse rate between 850- and 700-hPa levels |
| $LR_{850-500}$ | °C km$^{-1}$ | Environmental | Temperature lapse rate between 850- and 500-hPa levels |
| $LR_{700-500}$ | °C km$^{-1}$ | Environmental | Temperature lapse rate between 700- and 500-hPa levels |
| $T_{d,press}$ | °C | Environmental | Mean dew point temperature below 925-hPa level |
| $T_{d,alt}$ | °C | Environmental | Mean dew point temperature below 1 km AGL |
| CLW | kg m$^{-2}$ | Convective | AMPR-derived columnar cloud liquid water path |
| $PCT_{10}$ | K | Convective | AMPR 10.7-GHz polarization-corrected temperature |
| $PCT_{19}$ | K | Convective | AMPR 19.35-GHz polarization-corrected temperature |
| $PCT_{37}$ | K | Convective | AMPR 37.1-GHz polarization-corrected temperature |
| $PCT_{85}$ | K | Convective | AMPR 85.5-GHz polarization-corrected temperature |
| $Z_{95,Ku}$ | dBZ | Convective | APR-3 Ku-band 95th percentile composite reflectivity |
| $Pixels_{Ku}$ | unitless | Convective | APR-3 Ku-band composite reflectivity pixels $\geq$ 30 dBZ |
| DFR | unitless | Convective | APR-3 Ku-/Ka-band dual-frequency ratio |
| $AOT_{355}$ | unitless | Aerosol | HSRL2 355-nm aerosol optical thickness |
| $AOT_{532}$ | unitless | Aerosol | HSRL2 532-nm aerosol optical thickness |
| $Ext_{355}$ | Mm$^{-1}$ | Aerosol | HSRL2 355-nm aerosol extinction |
| $Ext_{532}$ | Mm$^{-1}$ | Aerosol | HSRL2 532-nm aerosol extinction |
| $Bsc_{355}$ | Mm$^{-1}$ sr$^{-1}$ | Aerosol | HSRL2 355-nm aerosol backscatter |
| $Bsc_{532}$ | Mm$^{-1}$ sr$^{-1}$ | Aerosol | HSRL2 532-nm aerosol backscatter |

The LCL altitude indicates cloud-base height and is often used in forecasting convection (Markowski and Richardson, 2010), though the exact role of LCL altitude on convective intensity is debated in the literature (e.g., Mulholland et al., 2021; Grabowski, 2023). All CAPE and LCL values were calculated using functions within the Python programming language (i.e., May et al., 2022) as noted in the data availability statement. K-Index is used to forecast convective potential/prevalence (i.e., not intensity) and is defined as

$$\text{K-Index}(°C) = (T_{850} - T_{500}) + T_{d,850} - (T_{700} - T_{d,700}) \tag{3}$$

where $T_{850}$, $T_{700}$, and $T_{500}$ are temperatures at the 850-, 700-, and 500-hPa levels, respectively, and $T_{d,850}$ and $T_{d,700}$ are dew point temperatures at the 850- and 700-hPa levels, respectively (George, 1960). From Eq. (3), K-Index considers: 1) low-to-mid-level LR, 2) low-level $T_d$, and 3) mid-level $T_d$ depression, with the former two (latter one) being directly (inversely) related to convective potential. K-Index was calculated semi-manually by identifying the pressure array elements nearest the 850-, 700-, and 500-hPa levels, extracting the associated $T$ and/or $T_d$ values from these elements, and utilizing Eq. (3). In a similar manner, the temperature and altitude values from array elements nearest the 850-, 700-, and 500-hPa levels were used to calculate $LR_{850-700}$, $LR_{850-500}$, and $LR_{700-500}$ as

$$\text{LR}(\text{°C km}^{-1}) = -\frac{(T_{upper} - T_{lower})}{(z_{upper} - z_{lower})}, \tag{4}$$

where LR is lapse rate, $T_{upper}$ and $T_{lower}$ are temperatures at the higher and lower altitudes, respectively, and $z_{upper}$ and $z_{lower}$ are the higher and lower altitudes, respectively. In addition to 850–500-hPa, 700–500-hPa LR may serve as an excellent indicator of convective potential (e.g., Sherburn and Parker, 2014). Others (e.g., Wang et al., 2015) have used 850–700-hPa LR in forecasting convective potential due to its association with parcel vertical acceleration in the lower atmosphere. Lastly, low-level $T_d$ is important for convective intensity due to entrainment of relatively high-water-vapor air into an updraft's base (e.g., Lucas et al., 2000). Mean low-level $T_d$ values were calculated by finding array elements where 1) pressure was > 925 hPa, or 2) altitude was < 1 km AGL, and calculating mean $T_d$ from the associated array elements.

We utilize microwave remote-sensing signatures from radar and radiometer to evaluate convective intensity and prevalence. The 30-dBZ $Z_H$ isoline has often been used to identify precipitation regions (e.g., Straka et al., 2000) and delineate between different "storms" or "cells" (e.g., Johnson et al., 1998; Hastings and Richardson, 2016; Amiot et al., 2019). As precipitation-sized hydrometeors form and grow, $Z_H$ increases due to hydrometeor diameter weighting to the sixth power associated with Rayleigh scattering, with eventual onset of non-Rayleigh resonance effects for larger hydrometeor diameters relative to the radar wavelength (Rinehart, 2010). This is especially important to note at finer wavelengths, such as 2.2 and 0.84 cm associated with APR-3's Ku and Ka bands, respectively (Durden et al., 2020b), the primary radar dataset used herein. A combination of Ku- and Ka-band radar data can be powerful when evaluated using DFR:

$$\text{DFR} = Z_{Ku} - Z_{Ka}, \tag{5}$$

where $Z_{Ku}$ and $Z_{Ka}$ represent $Z_H$ at Ku- and Ka-band, respectively, on a logarithmic scale (i.e., expressed in dBZ) (e.g., Liao et al., 2008; Liao and Meneghini, 2011). In regions where $Z_{Ku}$ and $Z_{Ka}$ are both similar (e.g., near 0 dBZ for hydrometeors that are in the Rayleigh scattering regime at both frequencies), DFR will be near zero; however, departures in DFR from 0 dBZ can indicate differences in attenuation between the two frequencies and can be used to infer hydrometeor size and phase (e.g., Liao and Meneghini, 2011). As Ku-band $Z_H$ increases, the DFR in rain regions generally becomes slightly negative (i.e., -1–0) before increasing to positive values for $Z_H > 30$ dBZ; in regions of ice hydrometeors, DFR generally increases with increasing Ku-band $Z_H$ as seen in Eq. (5), with a steeper increase occurring for lower-density ice hydrometeors (Liao and Meneghini, 2011). In our study, DFR values < -3 or > 15 were masked to avoid regions where the Ku- or Ka-band data were severely attenuated (e.g., Durden et al., 2020a).

Microwave radiometers generally retrieve higher $T_b$ values at increasingly lower frequencies as precipitation hydrometeors grow in the absence of ice formation aloft (e.g., Spencer et al., 1994). This makes it possible to retrieve cloud and precipitation properties using $T_b$ combinations (e.g., Wilheit and Chang, 1980; Wentz and Spencer, 1998; Hong and Shin, 2013; Amiot et al., 2021). AMPR's CLW retrievals often fail within precipitation regions; thus, as a cloud grows vertically, AMPR-derived CLW is expected to increase until it fails in moderate-to-heavy precipitation (Amiot et al., 2021; Amiot, 2023). Because of this, we will focus more on AMPR polarization-corrected temperature (PCT) in section 3 than CLW, though a similar CLW analysis is included in supplemental material. However, CLW

increasing around precipitation may yield useful information about the associated convective intensity; for example, precipitation is often associated with cumulus clouds at least 1.5–2 km tall (Smalley and Rapp, 2020) and CLW > 1 kg m$^{-2}$ may indicate precipitation formation within these clouds (e.g., Jiang and Zipser, 2006).

One remote-sensing instrument employed in aerosol analyses is lidar, including HSRL2 for CAMP$^2$Ex (Hostetler, 2020; Reid et al., 2023; Ferrare et al., 2023). HSRL2 measures aerosol backscatter and depolarization ratio at 355, 532, and 1064 nm, with aerosol extinction and AOT also measured using the HSRL2 technique at 355 and 532 nm (Hostetler, 2020). Integration for calculating AOT occurs over a vertical distance starting near the surface and ending at the top of the aerosol extinction profile, which is often around 5–6 km AGL when deployed from the P-3. The top altitude is typically ~1.5 km below the aircraft altitude. Lenhardt et al., (2022) and Redemann and Gao (2024) demonstrated how HSRL2's extinction and backscatter coefficients, especially at 532 nm, have strong direct correlations with cloud condensation nuclei concentrations. Additional studies (e.g., Liu et al., 2016) noted a direct correlation between lidar-based AOT and cloud condensation nuclei concentration. Therefore, extinction, backscatter, and AOT may all be considered when examining aerosol concentration. However, the height/location of an aerosol layer, which can be obtained from extinction and/or backscatter, is important to consider when evaluating diabatic heating from radiation absorption (e.g., Chand et al., 2009; Redemann et al., 2021).

### 2.4 Data matching in CAMP$^2$Ex scenes

Once the above parameters were calculated from each dropsonde throughout CAMP$^2$Ex P-3 science flights 05–19, they were matched spatiotemporally with APR-3 and AMPR data. AMPR was inoperable during science flight 01 and had un-optimized settings for its gain and offset values during science flights 02–04 (Lang et al., 2021), resulting in the exclusion of science flights 01–04 at the outset of our study. A "scene" was then established for each dropsonde, defined herein using a standard duration of 10 minutes, calculated as ±5 minutes from the dropsonde launch time. The APR-3 scans nearest the start and end of this time window were identified and, to account for situations where radar data collection began shortly before or after the start and/or end time of a given scene (e.g., P-3 was turning at the calculated start or end time), a grace of ±1 minute was allowed for the total scene duration, yielding a 10% uncertainty in scene duration. Scenes where the time difference between the APR-3 scans nearest the start and end of the scene was < 9 or > 11 minutes (e.g., due to significant aircraft maneuvers at the start and/or end of the scene) were masked from the analysis, which amounted to 47 dropsondes in total. Out of the 144 initial dropsondes, five were removed due to the aforementioned relative humidity analysis, and the removal of these 47 additional dropsondes yielded a total of 92 dropsondes retained for our study. Infrequently, applying the data masks discussed in section 2.2 also resulted in the masking of all AMPR and/or APR-3 data in a given scene. The AMPR, APR-3, and HSRL2 scans nearest the start and end times of each scene were noted, and all AMPR, APR-3, and HSRL2 data were examined over the same approximate time period within each scene; an example of these data in a single scene is provided in Fig. 1.

Eight remote-sensing parameters related to convective intensity and/or prevalence were calculated in each scene: 95th percentile (p95) of AMPR PCT at 10.7, 19.35, 37.1, and 85.5 GHz; p95 of AMPR CLW, p95 of APR-3 Ku-band

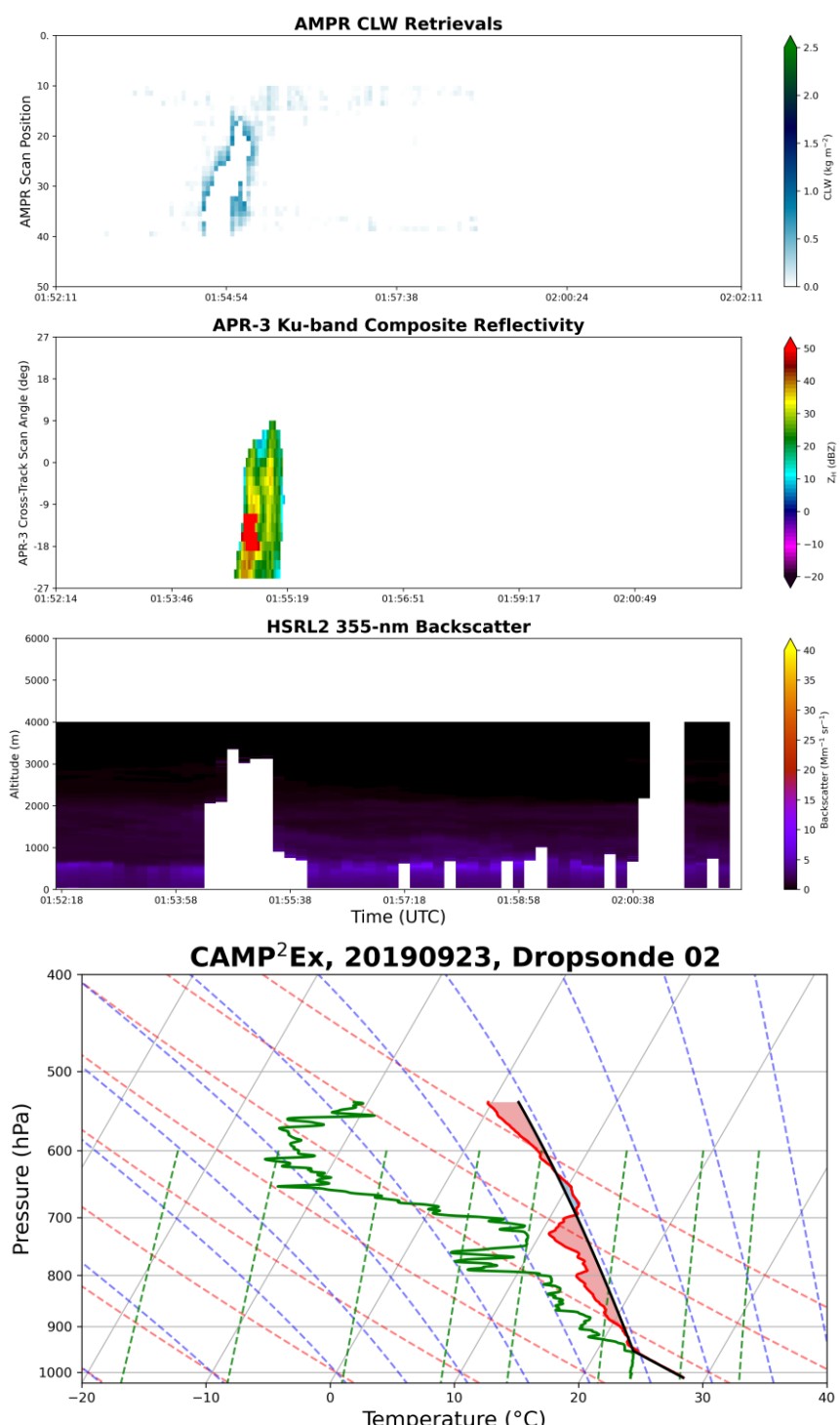

**Figure 1: Strip charts (i.e., top-view time series) of AMPR CLW (top) and APR-3 Ku-band composite $Z_H$ (second from top) along with a time-height plot of HSRL2 532-nm backscatter (second from bottom) during approximately 0152 to 0202 UTC on 24 Sept 2019 (23 Sept 2019 flight date based on takeoff time), which cover the "scene" associated with the dropsonde launched at 0157:15 UTC shown in the skew-T log-P diagram (bottom panel). All AMPR data flags have been applied in the top panel. Red shading in the bottom panel indicates CAPE, while solid red, green, and black lines denote temperature, dew point temperature, and parcel temperature, respectively.**

composite $Z_H$ ($Z_{95,Ku}$) and DFR, and the number of APR-3 Ku-band composite $Z_H$ pixels $\geq$ 30 dBZ (Pixels$_{Ku}$), hereafter referred to by their symbols in Table 1. The p95 values were used for the former seven parameters due to their direct association with peak convective intensity (e.g., increased raindrop size and radar reflectivity with stronger updraft, all else being equal; e.g., Kollias et al., 2001), with the 95th percentile employed to avoid potential outlier values associated with maximum values. Ku-band was used for the composite $Z_H$ analyses given its reduced attenuation compared to a Ka-band signal over the same distance and atmospheric conditions (i.e., all else being equal). To calculate composite $Z_H$, the data quality control described in Amiot (2023) was applied to all 25 APR-3 scan angles in each scene. Within each column of quality-controlled APR-3 data across science flights 05–19, the maximum $Z_H$ between the P-3 altitude and the surface was used as the composite $Z_H$. The presence of occasional residual near-surface range-/sidelobe effects at off-nadir scan angles was noted, which often manifested as very high composite $Z_H$ (i.e., > 70 dBZ). As a basic restriction, all composite $Z_H$ pixels > 70 dBZ were excluded from our analyses, but some erroneous pixels may still reside in the final dataset (e.g., isolated cases with some noisy pixels and/or near-surface range-/sidelobe effects with $Z_H$ < 70 dBZ). Once all composite $Z_H$ values were calculated, $Z_{95,Ku}$, DFR, and Pixels$_{Ku}$ were recorded in each scene. AMPR PCT values were calculated following the methods of Cecil and Chronis (2018), with their methods for 89.0-GHz data applied directly to AMPR's 85.5-GHz data. The p95 PCT in each AMPR channel was recorded along with the p95 of retrieved AMPR CLW in each scene.

## 2.5 Environmental stratification and sensitivity testing

To begin isolating potential aerosol influences on tropical convection, two steps were employed: 1) bin the environmental scenes into different groups based on a particular AVAPS parameter and magnitude, and 2) incorporate HSRL2 data into this analysis. The AVAPS parameters, which are the eight "environmental" parameters in Table 1, were utilized. To stratify each environment, a single AVAPS parameter was separated into "low," "medium," and "high" values, and each scene was grouped into one of these categories based on the associated dropsonde's values. Within each environmental bin, the eight convective parameters (Table 1) were compared against mean values of the six HSRL2 parameters, which are the "aerosol" parameters in Table 1, from each scene. The main statistics examined were Pearson correlation coefficients, the number of data points used in each comparison, and the statistical significance, primarily based on whether the p-value associated with the Pearson correlation coefficient was < 0.01 (e.g., Wilks, 2011). A few subjectively selected correlations were examined in greater detail using scatterplots, wherein it should be noted that the exact number of data points varied from plot-to-plot due to variations in missing data (e.g., dropsonde launched below the 500-hPa level for any parameters that use 500-hPa data).

To further mitigate potential impacts of any outliers, a bootstrapping approach was utilized when examining the Pearson correlation coefficient, number of data points, and p-value in each convective-aerosol parameter pairing within each bin for each environmental parameter. To apply the bootstrapping method, 10% of the convective-aerosol data pairs were withheld randomly before calculating the Pearson correlation coefficient and associated p-value for the remaining number of data points. This process was repeated 1000 times for each convective-aerosol pairing, and mean values of the Pearson correlation coefficients, number of data points, and p-values were calculated; these mean

values are reported throughout section 3 below. Though 1000 was often much greater than the number of unique combinations possible for a given data pairing, 1000 resamples for a bootstrapping approach is common in the literature and was applied uniformly across all data pairs in this study. Comparisons involving a sample size $< 10$ were unchanged by this approach of withholding 10% of the dataset since the remaining sample size was rounded up to the nearest whole integer, which always returned the original number of data points if the sample size was $< 10$.

Lastly, the exact values used to stratify each environmental condition were varied in a sensitivity test consisting of four different sets of thresholds for each parameter (Table 2). The methods used to stratify the environmental parameters in Tests 1–4 were, respectively, as follows:

1) Create campaign-wide histograms of the AVAPS parameter and visually identify approximate values that split the dataset into three roughly equal-sized groups.

2) Objectively select thresholds that split each parameter's dataset into three equal-sized groups (see the Data Availability statement).

3) Manually select thresholds that fall between the low-medium and medium-high thresholds previously identified in Tests 1 and 2.

4) Objectively select thresholds that split each parameter's dataset into three groups where the "low" and "high" categories each contain 25% of the data and the "medium" category contains 50% of the data (i.e., "medium" datasets that were approximately twice as large as the "low" and "high" datasets).

For brevity, only results from Test 2 are shown herein, but results from all four tests can be found in supplemental material. Test 2 is highlighted due to its objective stratification into roughly equal-sized groups. The bootstrapping method described above was also applied in all sensitivity tests.

Relating the parameters discussed in this section back to our hypotheses listed at the end of section 1, we hypothesized that PCT, $Z_{95,Ku}$, Pixels$_{Ku}$, and DFR would all increase under greater NCAPE$_{mod}$, K-Index, LRs, and low-level $T_d$, though, again, this investigation is secondary to our aerosol analyses in this study as outlined at the end of section 1. Expectations for LCL altitude were more uncertain given some differences in the results and discussions in past works that have explored LCL altitude related to convective intensity (e.g., Mulholland et al., 2021; Grabowski, 2023).

### 3.  AMPR results

This section presents the results of comparing the AMPR-based convective parameters with HSRL2 data within environmental bins established using the eight AVAPS parameters. Correlation tables are used to provide complete descriptions of the observed correlations, with more in-depth discussions and analyses performed for some subjectively selected correlations that were statistically significant and/or potentially most impactful. Brief descriptions of the sensitivity test results are provided, and all associated correlation tables from these sensitivity tests can be found in supplemental material.

**Table 2: List of the four sensitivity tests that were performed to stratify the eight AVAPS parameters into "low," "medium," and "high" bins. The listed values in each bracket represent the inclusive range of the "medium" bin for the respective parameter and test; that is, values less (greater) than the lower (upper) limit were classified into the "low" ("high") bin.**

| AVAPS parameters | Test 1: Visual histogram analysis | Test 2: Objective split 0.33-0.33-0.33 | Test 3: Manual selection between Tests 1 and 2 | Test 4: Objective split 0.25-0.50-0.25 |
|---|---|---|---|---|
| $T_{d,alt}$ | [21.0, 22.5] °C | [21.72, 22.4] °C | [21.35, 22.45] °C | [21.52, 22.59] °C |
| $T_{d,press}$ | [22.0, 23.0] °C | [22.62, 23.2] °C | [22.3, 23.1] °C | [22.34, 23.39] °C |
| $LR_{700-500}$ | [5.5, 6.0] °C km$^{-1}$ | [5.52, 5.9] °C km$^{-1}$ | [5.51, 5.95]°C km$^{-1}$ | [5.39, 6.01] °C km$^{-1}$ |
| $LR_{850-500}$ | [5.0, 5.5] °C km$^{-1}$ | [5.18, 5.43] °C km$^{-1}$ | [5.1, 5.47] °C km$^{-1}$ | [5.12, 5.46] °C km$^{-1}$ |
| $LR_{850-700}$ | [4.5, 5.5] °C km$^{-1}$ | [4.25, 4.98] °C km$^{-1}$ | [4.35, 5.25] °C km$^{-1}$ | [4.06, 5.11] °C km$^{-1}$ |
| K-Index | [30, 35] °C | [31.08, 35.61] °C | [30.5, 35.3] °C | [30.07, 36.59] °C |
| LCL | [400, 550] m | [404.1, 480.28] m | [402, 525] m | [369.36, 509.86] m |
| $NCAPE_{mod}$ | [0.04, 0.06] m s$^{-2}$ | [0.03, 0.05] m s$^{-2}$ | [0.035, 0.055] m s$^{-2}$ | [0.02, 0.06] m s$^{-2}$ |

To investigate AMPR data retrieved in and around precipitation, AMPR $T_b$ values can be used to obtain PCTs in these regions. Correlation coefficients between AMPR's 19.35-GHz PCT and the HSRL2 parameters are shown in Fig. 2. For brevity, only $PCT_{19}$ is detailed herein given its sensitivity to clouds and precipitation, with additional PCT results presented in supplemental material. From Fig. 2 some positive correlations were present and statistically significant, which illustrates the benefits of examining PCT in regions of precipitation wherein AMPR CLW retrievals may fail, as discussed below and in supplemental material. However, most of the correlations throughout Fig. 2 were fairly weak and had limited statistical significance, especially when comparing $PCT_{19}$ with the extinction parameters which indicates a level of inconclusiveness that remains in these results. Despite this, some of the correlations with greatest positive statistical significance were examined in more detail using scatterplots, in particular $PCT_{19}$ versus $Bsc_{355}$ when binned by LCL (Fig. 3a) and $PCT_{19}$ versus $Bsc_{355}$ when binned by $LR_{850-500}$ (Fig. 3b).

When examining the $PCT_{19}$ values in Fig. 3a, many data points are < 200 K, which indicates a relative lack of considerable precipitation in the scenes examined (Amiot et al., 2021; Amiot 2023); however, several data points with $PCT_{19}$ > 200 K can be seen in Fig. 3a, including values > 260 K, which indicates that $PCT_{19}$ is indeed capturing precipitation. There is considerable clustering of the data between a $PCT_{19}$ of 185–200 K and $Bsc_{355}$ of 0–4.5 Mm$^{-1}$ sr$^{-1}$, suggesting the presence of several instances of clouds that were generally not precipitating. The association of the highest $PCT_{19}$ values with relatively low aerosol concentrations (i.e., < 3 Mm$^{-1}$ sr$^{-1}$ in this case) within the low and medium LCL groups, combined with the clustering of data points mentioned previously, seems to have caused the extremely low correlation values for these low and medium groups. It appears that the high-LCL correlation was sensitive to the two outlier values with $PCT_{19}$ > 220 K and $Bsc_{355}$ > 5 Mm$^{-1}$ sr$^{-1}$, which contributed to its high value; however, the fact that this correlation was found to have statistical significance indicates that it is worthy of further examination. In general, increased aerosol concentration was not strongly associated with enhanced convection in Fig. 3a. However, it is noteworthy that over half of the data points with $PCT_{19}$ > 240 K were associated with a low LCL; this would indicate a relatively high amount of low-level water vapor content, wherein warm-phase convective invigoration may take place (e.g., Grabowski and Morrison, 2020); however, there are only seven data points with $PCT_{19}$ > 240 K and the aerosol concentrations are relatively low, so this potential connection needs further analysis.

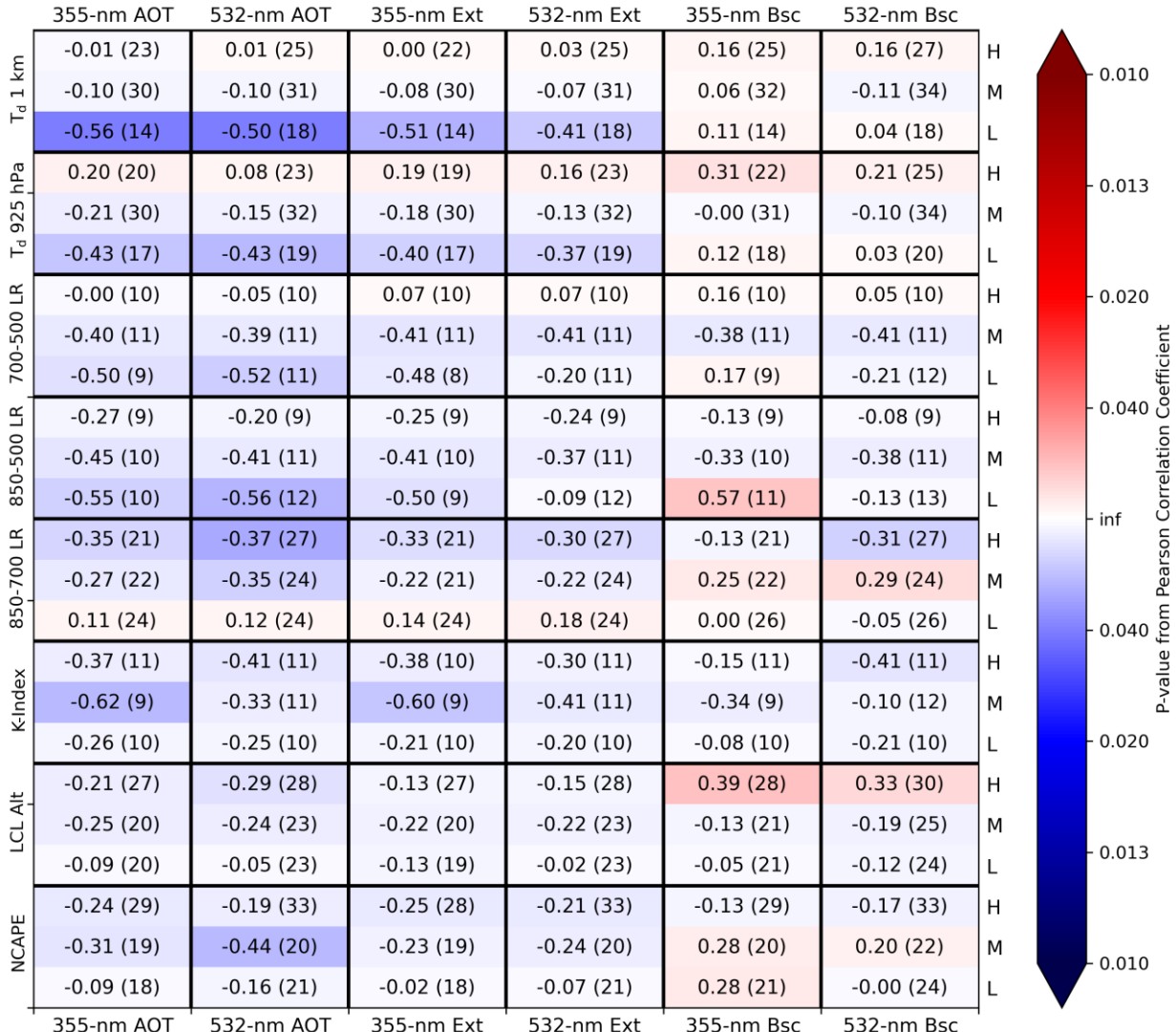

**Figure 2: Pearson correlation coefficients from comparing p95 AMPR 19.35-GHz PCT with mean HSRL2 AOT, extinction (Ext), and backscatter (Bsc) at 355 and 532 nm (top and bottom borders) within environmental bins stratified by the eight AVAPS parameters (left border) at low (L), medium (M), and high (H) magnitudes (right border) across the CAMP²Ex scenes. AVAPS magnitudes were stratified using the values of Test 2 (Table 2). Within each cell, the listed value is the Pearson correlation coefficient and the parenthesized value indicates the mean number of data points used in the (bootstrapped) comparison. Cells with a Pearson correlation coefficient ≥ 0.70 contain bolded text. Reds (blues) represent positive (negative) Pearson correlation coefficients, and the color shading corresponds to the magnitude of the p-value according to the colorbar, with darker shades of each color associated with lower p-values (i.e., greater statistical significance). Color shading begins to increase substantially around a p-value of 0.05 and reaches a maximum for p-values around 0.01.**

The impact of a reduced dataset size can be seen to an even greater degree in Fig. 3b, which contains far fewer data points compared to Fig. 3a due to $LR_{850-500}$ requiring data from the 500-hPa level. Despite this, a statistically significant positive correlation was found between the aerosol and convective metrics, but it was unexpected that this occurred for the low $LR_{850-500}$ group and not the medium or high groups. From Table 2, the low $LR_{850-500}$ values were still conditionally unstable and thus supportive of convection, so this result is physically plausible and deserves further

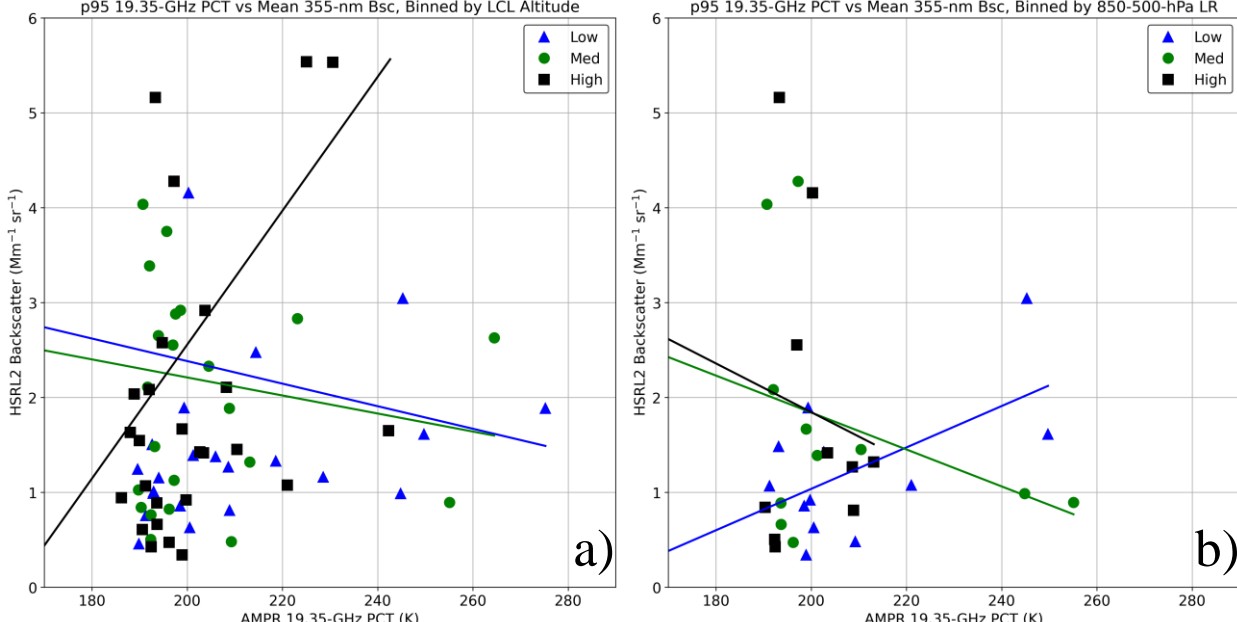

**Figure 3: Scatterplots of p95 PCT$_{19}$ compared with mean values of the HSRL2 parameter listed in the title and y-axis of each plot within environmental bins stratified using the AVAPS parameter listed in the title of the corresponding plot. AVAPS threshold values were from Test 2 (Table 2). In all plots, blue triangles, green circles, and black squares correspond to data points associated with low, medium, and high magnitudes, respectively, of the associated AVAPS parameter. Please note that the ranges of the x- and y-axes are not constant among the scatterplots shown.**

analysis. However, it seems that the data points with PCT$_{19}$ > 240 K greatly influenced the correlations for the low and medium groups, which is expected with a relatively limited sample size. The increase in PCT$_{19}$ with increasing aerosol concentration (within the low group) matches the hypothesized correlations, but the highest aerosol concentrations in Fig. 3b (i.e., Bsc$_{355}$ > 2 Mm$^{-1}$ sr$^{-1}$ in this case) were associated with relatively low PCT$_{19}$ values < 205 K in all but one instance. Thus, the overall trends in correlation between aerosol concentration and PCT$_{19}$ in Fig. 3b are fairly mixed.

Due to the presence of statistically significant and potentially impactful results in Figs. 2 and 3, future work should examine these and other AMPR data in greater detail. Several trends were consistent across the sensitivity tests (supplemental material). However, it should be acknowledged that many correlations throughout Fig. 2 were fairly weak (i.e., magnitude < 0.3) and/or had low statistical significance. Additionally, as will be referenced throughout these discussions, a relatively limited sample size was present for several of the comparisons/scatterplots, and all cases examined in this study would benefit from a larger sample (e.g., incorporating satellite and/or modeling data to expand the number of data points throughout the study domain, especially in and around deeper convection). Lastly, as is true for all analyses herein, while high correlation between two parameters is interesting and potentially significant, it does not guarantee a cause-and-effect situation. Thus, the most noteworthy trends identified in this study (e.g., Fig. 3) should be examined further to evaluate their significance and potential aerosol influences on convection.

AMPR's CLW comparisons with HSRL2 in the stratified environments are summarized via Figs. S1–S3 in supplemental material. Due to the CLW retrieval method not accounting for precipitation, regions wherein high CLW would be expected in association with heavy precipitation could easily appear as a region of failed retrieval or may return a high CLW value with an unknown uncertainty (Amiot et al., 2021). Since we focus on the strongest convection observed in this study, these uncertainties make it more difficult to assess AMPR CLW in relation to the study focus, which is why we have chosen to present this information in supplemental material. From Fig. S1, when low CLW values (i.e., $< 1$ g m$^{-2}$) were included in the dataset, many Pearson correlation coefficients between CLW and the aerosol parameters were negative and with statistical significance, especially when comparing CLW with AOT and Ext at either HSRL2 wavelength. This can be seen further in the scatterplots of CLW versus AOT$_{355}$ when binned by NCAPE$_{mod}$ (Fig. S2a) and CLW versus Bsc$_{532}$ when binned by LR$_{850-700}$ (Fig. S2b), wherein the broad distribution of AOT$_{355}$ and Bsc$_{532}$ values around a CLW value of 0 kg m$^{-2}$ greatly contributed to these negative correlations. Masking CLW $< 1$ g m$^{-2}$ removed much of the statistical significance throughout the correlations, as seen in Fig. S3. Many of the remaining correlations were negative, as seen in Figs. S2c, S2d, and S3, but these correlations were generally weak. Thus, despite the presence of some stronger and more statistically significant correlations in Figs. S1–S3, the abundance of weaker correlations indicates a considerable degree of inconclusiveness that remains in these results. Moreover, these trends persist across the sensitivity tests (supplemental material). This further emphasizes the difficulty in truly separating aerosol influences from environmental effects, in addition to the limitations associated with the observational datasets and the relatively small sample sizes employed in our study.

## 4. APR-3 results

Similar analyses are presented in this section using Z$_{95,Ku}$, Pixels$_{Ku}$, and DFR as the convective parameters. All figures utilize the AVAPS thresholds from Test 2 (Table 2), with the full sensitivity test results presented in supplemental material. To begin, Pearson correlation coefficients between Z$_{95,Ku}$ and the HSRL2 variables can be found in Fig. 4. Several statistically significant positive correlations can be seen between Z$_{95,Ku}$ and the HSRL2 variables in Fig. 4, most notably for some comparisons when binned according to NCAPE$_{mod}$, LR$_{850-700}$, and LR$_{700-500}$. However, most of the correlations throughout Fig. 4 were relatively weak and had limited statistical significance. Interestingly, many of the strongest correlations with the highest statistical significance (albeit still fairly limited) were often associated with the low category of a given environmental group. To examine some of these trends in greater detail, the two parameters selected for more in-depth analysis from Fig. 4 were: 1) Bsc$_{355}$ binned by LR$_{700-500}$ (Fig. 5a), and 2) Bsc$_{532}$ binned by NCAPE$_{mod}$ (Fig. 5b), which were selected due to their relatively high statistical significance in the low category of the environmental groups.

In Fig. 5a, many Z$_{95,Ku}$ values $\geq 30$ dBZ were present in the CAMP$^2$Ex scenes, indicating that precipitating systems were indeed overflown by the P-3 aircraft and further suggesting that AMPR's precipitation flags contributed to many of the unexpected negative correlations in section 3. The standout feature of Fig. 5a is the statistically significant positive correlation between Z$_{95,Ku}$ and Bsc$_{355}$ when binned by low LR$_{700-500}$ values. That is, as aerosol concentration

| | 355-nm AOT | 532-nm AOT | 355-nm Ext | 532-nm Ext | 355-nm Bsc | 532-nm Bsc | |
|---|---|---|---|---|---|---|---|
| **$T_d$ 1 km** | 0.12 (18) | 0.19 (18) | 0.11 (18) | 0.13 (18) | 0.31 (20) | 0.36 (20) | H |
| | -0.21 (23) | -0.05 (24) | -0.32 (23) | -0.25 (24) | -0.32 (25) | -0.29 (27) | M |
| | -0.15 (9) | -0.07 (11) | -0.13 (9) | -0.05 (11) | -0.10 (9) | -0.01 (11) | L |
| **$T_d$ 925 hPa** | 0.16 (17) | 0.15 (18) | 0.15 (16) | 0.15 (18) | 0.29 (18) | 0.34 (19) | H |
| | -0.14 (22) | 0.08 (23) | -0.28 (22) | -0.10 (23) | -0.30 (23) | -0.20 (25) | M |
| | -0.30 (12) | -0.24 (13) | -0.29 (12) | -0.30 (13) | -0.15 (13) | -0.16 (14) | L |
| **700-500 LR** | 0.38 (8) | 0.36 (8) | 0.46 (8) | 0.47 (8) | 0.41 (8) | 0.44 (8) | H |
| | -0.32 (10) | -0.23 (10) | -0.34 (10) | -0.34 (10) | -0.30 (10) | -0.16 (10) | M |
| | -0.32 (7) | -0.33 (7) | 0.04 (6) | 0.33 (7) | 0.65 (8) | 0.26 (8) | L |
| **850-500 LR** | -0.46 (8) | -0.40 (8) | -0.44 (8) | -0.43 (8) | -0.33 (8) | -0.22 (8) | H |
| | 0.34 (9) | 0.40 (9) | 0.23 (9) | 0.21 (9) | -0.11 (9) | 0.34 (9) | M |
| | 0.06 (9) | 0.11 (9) | 0.11 (8) | 0.16 (9) | 0.24 (9) | 0.41 (9) | L |
| **850-700 LR** | -0.45 (14) | -0.33 (17) | -0.44 (14) | -0.31 (17) | -0.33 (14) | -0.29 (18) | H |
| | -0.22 (18) | -0.17 (18) | -0.23 (17) | -0.25 (18) | 0.03 (18) | 0.02 (18) | M |
| | 0.40 (19) | 0.42 (19) | 0.27 (19) | 0.37 (19) | -0.22 (21) | -0.16 (21) | L |
| **K-Index** | -0.41 (9) | -0.36 (9) | -0.45 (9) | -0.34 (9) | -0.26 (9) | -0.30 (9) | H |
| | -0.33 (7) | -0.19 (7) | -0.18 (7) | -0.17 (7) | 0.40 (8) | 0.48 (8) | M |
| | 0.16 (9) | 0.30 (9) | 0.16 (9) | 0.18 (9) | 0.15 (9) | 0.31 (9) | L |
| **LCL Alt** | -0.27 (19) | -0.24 (19) | -0.26 (19) | -0.28 (19) | -0.10 (21) | -0.10 (21) | H |
| | -0.27 (13) | 0.08 (15) | -0.24 (13) | -0.09 (15) | 0.16 (14) | 0.01 (17) | M |
| | 0.35 (18) | 0.36 (19) | 0.05 (18) | 0.31 (19) | -0.25 (19) | -0.23 (20) | L |
| **NCAPE** | 0.01 (21) | 0.15 (22) | -0.08 (20) | 0.05 (22) | -0.26 (21) | -0.21 (22) | H |
| | -0.47 (17) | -0.29 (18) | -0.50 (17) | -0.38 (18) | -0.18 (18) | -0.07 (19) | M |
| | 0.42 (13) | 0.37 (14) | 0.33 (13) | 0.27 (14) | 0.38 (16) | 0.46 (17) | L |
| | 355-nm AOT | 532-nm AOT | 355-nm Ext | 532-nm Ext | 355-nm Bsc | 532-nm Bsc | |

**Figure 4: As in Fig. 2 but using p95 APR-3 Ku-band composite $Z_H$ as the convective parameter.**

increased within low-LR$_{700\text{-}500}$ conditions, the peak Ku-band $Z_H$ within the same scene increased as well, suggesting the development of larger raindrops. These large raindrops would dominate $Z_H$, but this analysis also highlights the importance of considering environmental conditions. Further, this comparison is hindered by the limited sample size, and thus a few data points may significantly impact the correlations. While the general trend of the LR$_{700\text{-}500}$ comparisons is similar across the sensitivity tests (supplemental material), the magnitude of the correlation and its statistical significance varied depending on the values used to stratify the lapse rates. Further, the highest $Z_{95,Ku}$ values were associated with relatively low values of Bsc$_{355}$ < 2 Mm$^{-1}$ sr$^{-1}$, suggesting that the largest raindrops were not necessarily correlated with the highest aerosol concentrations. Despite this, the positive correlations observed in this comparison warrant further exploration in future work.

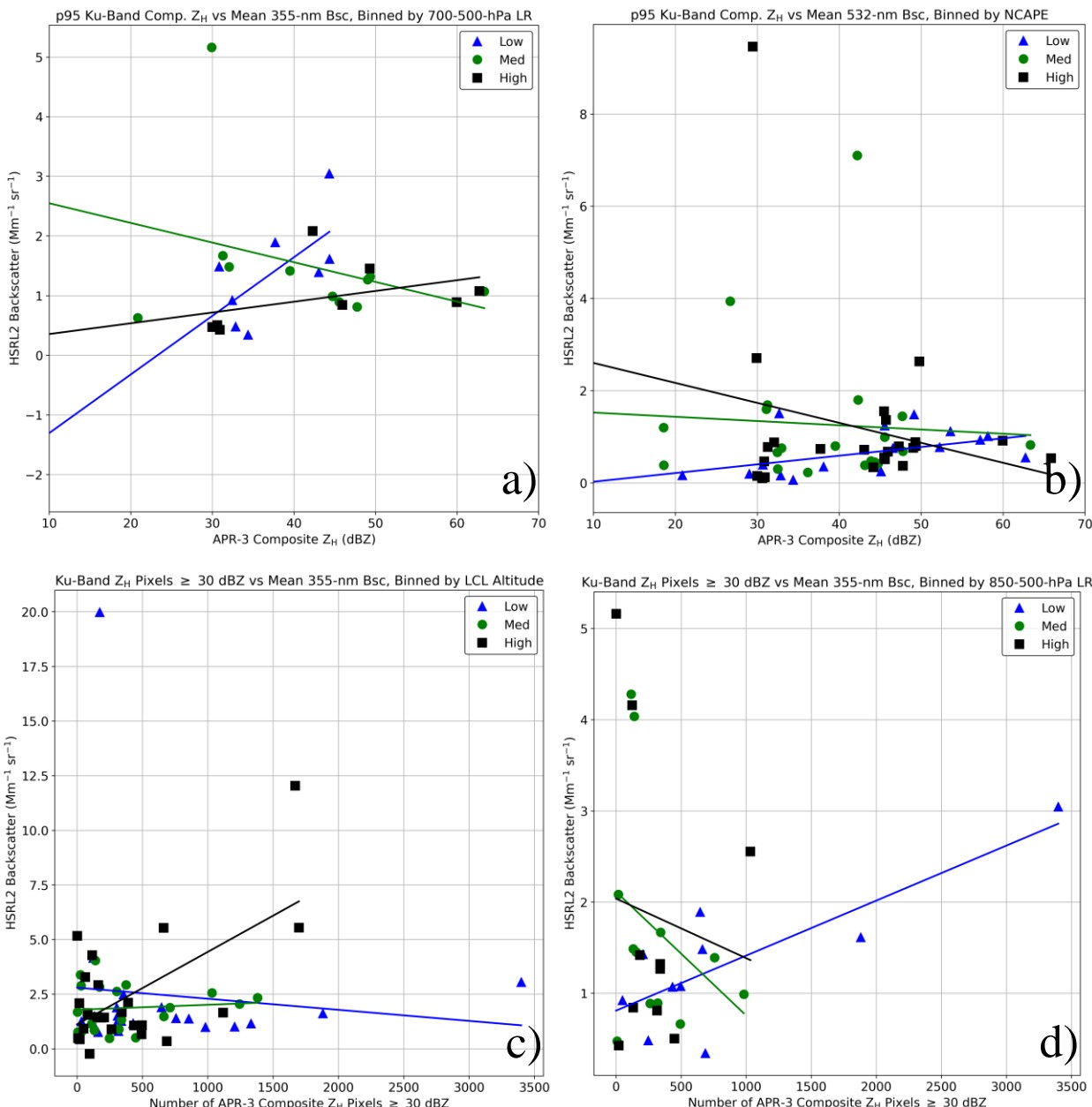

**Figure 5: As in Fig. 3, but these are scatterplots of p95 APR-3 Ku-band composite $Z_H$ (top row) and the number of APR-3 Ku-band composite $Z_H$ pixels $\geq$ 30 dBZ (bottom row) compared against the mean value of the HSRL2 parameter listed in the title of each plot. The AVAPS parameter used to stratify the environments is also listed in the title of each plot.**

Comparing $Z_{95,Ku}$ with $Bsc_{532}$ when binned by $NCAPE_{mod}$ (Fig. 5b), a positive correlation between $Z_{95,Ku}$ and $Bsc_{532}$ was present for the low $NCAPE_{mod}$ group, which matches expectations. However, the negative correlations for the medium and high $NCAPE_{mod}$ groups were interesting to see and seem to have been influenced by the $Bsc_{532}$ values > 2 $Mm^{-1}$ $sr^{-1}$, all of which were associated with $Z_{95,Ku}$ < 50 dBZ. The scattered data points with $Bsc_{532}$ < 2 $Mm^{-1}$ $sr^{-1}$ across the observed range of $Z_{95,Ku}$ indicates some inconclusiveness with these results, as higher $Z_{95,Ku}$ was not

necessarily observed with higher $Bsc_{532}$. As noted previously, these comparisons would benefit from additional analyses using a larger sample size that incorporates satellite and/or modeling data throughout the study domain.

Next, the number of APR-3 Ku-band composite $Z_H$ pixels $\geq$ 30 dBZ (i.e., $Pixels_{Ku}$) was used as the convective parameter (Fig. 6). Several more strongly positive and statistically significant correlations were present compared to Figs. 2 and 4, likely due to $Pixels_{Ku}$ focusing on the abundance of convection rather than a peak value in a given scene. The strongest positive correlations with a statistically significant p-value were found between $Pixels_{Ku}$ and lapse rate, K-Index, and LCL, which suggest some potentially interesting trends that warrant further exploration. However, considerable variability in the results can be seen across Fig. 6 as evidenced by the presence of many fairly weak correlations with limited statistical significance. Some correlations varied considerably depending on which aerosol parameter was examined (e.g., correlations with $AOT_{532}$ were moderately negative in the low $LR_{850-500}$ group but correlations with $Bsc_{355}$ were moderately positive in the same group). Several factors contribute to these differences and, as discussed previously, are difficult to account for entirely. Given their statistical significance, the decision was made to examine the relatively strong correlations between $Pixels_{Ku}$ and $Bsc_{355}$ within environments binned by LCL (Fig. 5c) and $LR_{850-500}$ (Fig. 5d). From Fig. 5c, most scenes featured at least one Ku-band composite $Z_H$ observation $\geq$ 30 dBZ, further indicating the precipitating clouds the P-3 passed over during CAMP²Ex. In Fig. 5c, with the available data, the general trend was an increase in $Pixels_{Ku}$ with increasing $Bsc_{355}$ for a higher LCL, regardless of whether these APR-3 pixels were part of a single large convective storm or several individual plumes. However, this trend appears to have been greatly influenced by the two "high" values with $Pixels_{Ku} > 1500$, which further suggests the limitations associated with using a relatively small dataset and would benefit from expansion in future work. The best-fit line for the low LCL group also appears to have been influenced by the $Pixels_{Ku}$ values $> 1500$. Mixed results from this comparison are further indicated by the highest aerosol concentration, around 20 $Mm^{-1}$ $sr^{-1}$, being associated with a low value of $Pixels_{Ku}$, $< 250$.

When comparing $Pixels_{Ku}$ with $Bsc_{355}$ in environments binned by $LR_{850-500}$, the limitations of the reduced sample size can be seen further, particularly for the low $LR_{850-500}$ group (Fig. 5d). Most notably, the correlation of 0.78 within the low group appears to be heavily influenced by the two data points with $Pixels_{Ku} > 1500$. Thus, while a strong and statistically significant correlation exists between $Pixels_{Ku}$ and $Bsc_{355}$ within environments of low $LR_{850-500}$, this relationship should be examined in more detail. This result was also unexpected, as greater conditional instability associated with higher lapse rates would generally indicate greater convective potential. In Fig. 5d, the results from the medium and high groups were scattered with $Pixels_{Ku} < 1100$, and the highest aerosol concentrations were associated with some of the lowest $Pixels_{Ku}$ values. Despite the limited sample in Fig. 5d, it was interesting to see cases such as this where the convective parameter was maximized for relatively medium aerosol concentrations, which indicates the potential for a "Goldilocks" zone of aerosol concentration enhancing convection.

Lastly, DFR was used as the convective metric (Fig. 7). As with $Z_{95,Ku}$ and unlike $Pixels_{Ku}$, DFR focuses on the intensity of a given convective storm rather than the overall abundance of convection. From Fig. 7, many statistically

| | 355-nm AOT | 532-nm AOT | 355-nm Ext | 532-nm Ext | 355-nm Bsc | 532-nm Bsc | |
|---|---|---|---|---|---|---|---|
| Td 1 km | -0.09 (23) | 0.00 (23) | -0.01 (22) | -0.03 (23) | 0.26 (25) | 0.06 (25) | H |
| | -0.23 (23) | -0.15 (24) | -0.28 (23) | -0.15 (24) | 0.01 (25) | -0.19 (26) | M |
| | -0.56 (11) | -0.56 (13) | -0.47 (11) | -0.47 (13) | 0.35 (11) | 0.25 (13) | L |
| Td 925 hPa | 0.03 (20) | -0.14 (21) | 0.15 (19) | 0.00 (21) | 0.41 (22) | 0.03 (23) | H |
| | -0.27 (24) | -0.17 (24) | -0.29 (24) | -0.18 (24) | 0.01 (25) | -0.17 (25) | M |
| | -0.53 (13) | -0.46 (15) | -0.40 (13) | -0.40 (15) | 0.38 (14) | 0.28 (16) | L |
| 700-500 LR | -0.28 (10) | -0.34 (10) | -0.21 (10) | -0.20 (10) | 0.13 (10) | -0.11 (10) | H |
| | -0.50 (10) | -0.52 (10) | -0.51 (10) | -0.51 (10) | -0.60 (10) | -0.61 (10) | M |
| | -0.35 (9) | -0.43 (9) | -0.26 (8) | -0.07 (9) | 0.33 (9) | -0.17 (10) | L |
| 850-500 LR | -0.41 (9) | -0.46 (9) | -0.40 (9) | -0.39 (9) | -0.13 (9) | -0.27 (9) | H |
| | -0.40 (11) | -0.34 (11) | -0.36 (11) | -0.35 (11) | -0.33 (11) | -0.39 (11) | M |
| | -0.48 (9) | -0.60 (9) | -0.34 (8) | 0.13 (9) | **0.78 (9)** | -0.07 (10) | L |
| 850-700 LR | -0.43 (16) | -0.54 (18) | -0.37 (16) | -0.34 (18) | -0.00 (16) | -0.46 (18) | H |
| | -0.43 (19) | -0.50 (20) | -0.28 (18) | -0.36 (20) | 0.56 (19) | 0.48 (20) | M |
| | 0.26 (22) | 0.32 (22) | 0.22 (22) | 0.25 (22) | -0.06 (24) | -0.10 (24) | L |
| K-Index | -0.38 (11) | -0.44 (11) | -0.36 (10) | -0.27 (11) | -0.02 (11) | -0.41 (11) | H |
| | -0.21 (9) | -0.03 (9) | -0.16 (9) | -0.06 (9) | -0.24 (9) | -0.19 (10) | M |
| | -0.15 (9) | -0.19 (9) | -0.14 (9) | -0.14 (9) | -0.23 (9) | -0.30 (9) | L |
| LCL Alt | -0.26 (21) | -0.47 (22) | -0.12 (21) | -0.20 (22) | 0.60 (23) | 0.49 (24) | H |
| | -0.22 (18) | -0.09 (18) | -0.21 (18) | -0.23 (18) | 0.09 (18) | -0.26 (19) | M |
| | -0.16 (18) | -0.08 (19) | -0.21 (18) | -0.01 (19) | -0.08 (19) | -0.23 (20) | L |
| NCAPE | -0.23 (27) | -0.18 (27) | -0.21 (26) | -0.18 (27) | -0.08 (27) | -0.17 (27) | H |
| | -0.49 (17) | -0.47 (17) | -0.34 (17) | -0.33 (17) | 0.36 (18) | 0.12 (18) | M |
| | -0.17 (14) | -0.03 (16) | -0.16 (14) | -0.13 (16) | 0.38 (17) | 0.01 (18) | L |
| | 355-nm AOT | 532-nm AOT | 355-nm Ext | 532-nm Ext | 355-nm Bsc | 532-nm Bsc | |

**Figure 6: As in Fig. 2 but using the number of APR-3 Ku-band composite $Z_H$ pixels ≥ 30 dBZ as the convective parameter.**

significant and strong correlations were associated with several of the environmental parameters but, unexpectedly, were often found within low values of these environmental conditions. While there was greater presence of statistically significant positive correlations in Fig. 7 compared to some of the other correlation analyses in this study, many of the correlations were relatively weak and/or statistically insignificant in these DFR analyses as well. To provide a detailed example of a convective-aerosol pair wherein the correlations were relatively weak for all environmental thresholds, we examined DFR compared with $Bsc_{355}$ when binned by $LR_{850-700}$ (Fig. 8a). That is, in Fig. 8a, all three environmental groups were associated with weakly negative or weakly positive correlations between $LR_{850-700}$ and $Bsc_{355}$. This resulted from most of the aerosol backscatter values falling within the range of 0–5 $Mm^{-1}$ $sr^{-1}$ while also covering nearly the full range of (masked) DFR values observed (i.e., near 0 to 15). The few data points with $Bsc_{355} > 5$ $Mm^{-1}$ $sr^{-1}$ were mainly associated with DFR < 10, which contrasts our hypothesis that greater Ka-band

**Figure 7: As in Fig. 2 but using p95 Ku-/Ka-band DFR as the convective parameter.**

attenuation would be associated with higher aerosol concentrations. The clustering of $Bsc_{355}$ between 2.5–5 $Mm^{-1} sr^{-1}$ with DFR between 12 and 15 is interesting and is more closely in line with the anticipated increase in attenuation, which warrants further investigation especially given the relatively small number of data points within this cluster.

Due to the presence of statistically significant correlations associated with $Bsc_{532}$ when binned by $NCAPE_{mod}$, we also examined it in greater detail (Fig. 8b). Some similar trends can be seen in Fig. 8b as in Fig. 8a, such as the negative correlations that occurred in scenes with medium values of the environmental parameter (i.e., $NCAPE_{mod}$). This was largely due to the clustering of data points with $Bsc_{532}$ of 0–2 $Mm^{-1} sr^{-1}$ across the full range of DFR values while the highest $Bsc_{532}$ values were associated with DFR < 10. The statistically significant positive correlation within the low $NCAPE_{mod}$ group follows expectations that greater attenuation would occur within stronger convection, but this comparison benefits from the relatively limited range of $Bsc_{532}$ observed within the low $NCAPE_{mod}$ group (i.e., $Bsc_{532}$

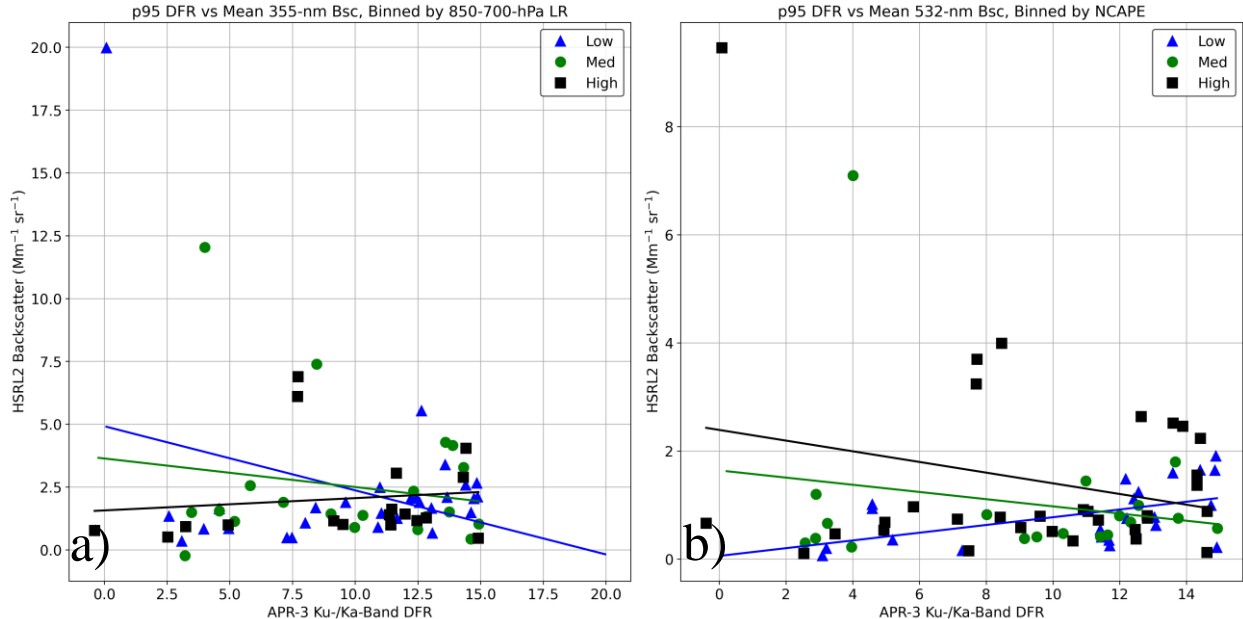

Figure 8: As in Fig. 3, but these are scatterplots of p95 APR-3 Ku-/Ka-band DFR compared against the mean
value of the HSRL2 parameter listed in the title of each plot. The AVAPS parameter used to stratify the
environments is also listed in the title of each plot.

of 0–2 Mm$^{-1}$ sr$^{-1}$). The trends in DFR compared to the aerosol parameters were also fairly consistent across the
sensitivity tests (supplemental material).

## 5.    Summary, limitations, and future work

This study focused on examining potential impacts of aerosol concentration on maritime tropical convection using
remote-sensing data in environmental contexts. Eight parameters from 92 AVAPS dropsondes across CAMP$^2$Ex P-3
science flights 05–19 were used to stratify the environments: modified normalized CAPE; LCL altitude; K-Index;
850–700-, 850–500-, and 700–500-hPa temperature lapse rates; mean $T_d$ below 1 km AGL; and mean $T_d$ below 925
hPa. Each dropsonde was associated with a 10-minute "scene," ±5 minutes from its launch time, wherein all aerosol-
convective comparisons were performed in relation to the given dropsonde-derived environment. Threshold values
were selected to divide scenes into "low," "medium," and "high" groups based on each AVAPS parameter, and
sensitivity testing examined four different sets of threshold values used for each stratification. Eight AMPR and APR-
3 metrics related to convective intensity and/or prevalence were compared with HSRL2 backscatter, extinction, and
AOT at 355 and 532 nm within the binned environments using Pearson correlation coefficients and their associated
p-values. These convective parameters were: p95 of PCT at 10.7, 19.35, 37.1, and 85.5 GHz; p95 of AMPR CLW,
p95 of APR-3 Ku-band composite $Z_H$; number of Ku-band composite $Z_H$ pixels ≥ 30 dBZ; and Ku-/Ka-band DFR.

The correlations between the convective metrics and aerosol parameters varied depending on which convective metric
was examined. During AMPR's CLW analyses, several negative correlations were observed, likely owing at least in

part to the precipitation masking and the presence of very low CLW values $< 1$ g m$^{-2}$, which limited the aerosol-convection conclusions that could be drawn solely based on AMPR CLW. This was mitigated when examining AMPR PCT$_{19}$, which includes precipitation regions and yielded several positive aerosol-convection correlations as expected. Likewise, examining APR-3's Z$_{95,Ku}$, Pixels$_{Ku}$, and DFR yielded many positive correlations with aerosol concentration, including several with statistical significance. Some of these strongest and most statistically significant correlations are potentially impactful and warrant exploration in future work. In particular, several noteworthy trends were observed when stratifying environments based on their temperature lapse rate or NCAPE$_{mod}$ which, among other environmental factors, suggests their importance to consider when examining aerosol-cloud effects. Some of the cases evaluated in more detail using scatterplots indicated the presence of a "Goldilocks" zone of medium aerosol concentration, suggesting that these medium values enhanced convection to a stronger degree than low or high aerosol concentrations in some cases, as observed in prior studies, while higher aerosol concentrations were associated with stronger convection in other cases, which matches our hypotheses. Many trends were also consistent across the sensitivity tests we employed. However, the results generally remain inconclusive as a result of widespread cases for each aerosol-convective analysis where the correlations within many of the environmental bins were weak and/or had limited statistical significance. This results largely from limitations in the sample used and limitations in the methods used to examine these observed data. The inconclusive results also highlight the difficulties in separating aerosol effects from environmental influences when solely using observational data.

These results are important as they highlight some potentially impactful correlations between convective parameters and aerosol concentrations in the maritime tropics, including some instances where medium-to-high aerosol concentrations appeared supportive of convective invigoration, while also demonstrating significant difficulties in separating environmental effects from aerosol effects. However, as noted throughout the manuscript, correlation does not necessarily indicate causation. Because of this, correlations highlighted in this study serve to identify potentially interesting and impactful trends that warrant a more in-depth exploration in future work, rather than providing solid definitive conclusions on their own. Further, the correlation tables presented in this manuscript, including those in supplemental material, provide a wide range of information that is applicable to broader applications (e.g., a future study that might explore the impacts of low-level $T_d$ or mid-level lapse rates on tropical convection).

While many results were encouraging, several limitations must be considered. Dropsondes launched when the P-3 was above 500 hPa were relatively limited, reducing the sample size for all associated environmental parameters. Other limitations in the dataset, such as the P-3 avoiding the most intense convection during a given flight and environmental modification from nearby convection, impacted the results. There was some ambiguity regarding whether an increase in Pixels$_{Ku}$ was associated with a single updraft or multiple updrafts, which have different implications for convective intensity and prevalence. Lastly, while many correlations were strong and encouraging, they do not necessarily prove a cause-and-effect situation for their respective comparison, as previously discussed. Thus, it is not possible to say with certainty that increased aerosol concentrations enhanced convection in these

CAMP$^2$Ex scenes solely based on the correlations presented in this study, but rather the data suggest the possibility for aerosol enhancement of convection and further analyses would increase confidence in these results.

Given the intriguing nature of some comparisons in this study, while also considering the above limitations, future work would greatly benefit these science questions. Future efforts could look at addressing the limitations above, such as using an advanced $Z_H$ attenuation-correction method and distinguishing areas where Pixels$_{Ku}$ were adjacent or separated. Employing other datasets from the P-3 and Learjet-35 aircraft, in addition to incorporating numerical simulations, may increase the reliability of the strongest correlations observed. While our study sought to emphasize data from the P-3 aircraft, other remote-sensing data (e.g., satellite) may help with assessing nearby convection just outside of the P-3 observation range. Other convective metrics could supplement those selected for this study, such as peak 30-dBZ $Z_H$ contour height in a storm given its direct relation to updraft magnitude (e.g., Straka et al., 2000; Amiot et al., 2019). Likewise, additional environmental parameters, such as wet-bulb potential temperature profiles (Williams and Renno, 1993), would be useful to examine. We focused on aerosol concentrations given their significant influence on cloud particle size distributions (all else being equal), but other aerosol properties (e.g., type, composition, and hygroscopicity) and their vertical location/distribution may also be helpful to consider, as would direct computation of updraft vertical velocity. Lastly, as one of the main takeaways from this manuscript, examining some of the most-significant convective-aerosol correlations in greater detail would be of significant benefit.

**Data availability**

The AMPR, APR-3, AVAPS, and HSRL2 data are available on the NASA Langley Research Center's Airborne Science Data for Atmospheric Composition repository at https://www-air.larc.nasa.gov/cgi-bin/ArcView/camp2ex, cited herein as Aknan and Chen (2020). Objective selections of threshold values for environmental stratification were performed using Python's NumPy nanpercentile function, and the random data sampling during the bootstrapping approach was accomplished using Python's NumPy random.choice function (Harris et al., 2020). Pearson correlation coefficients and p-values were calculated using Python's SciPy pearsonr function (Virtanen et al., 2020). Several of the environmental parameters were calculated using Python's MetPy package (May et al., 2022), including the mixed_layer_cape_cin function for CAPE$_{mod}$, the calc.lcl function for LCL altitude, and SkewT function for producing the dropsonde image in Fig. 1.

**Author contributions**

CGA performed all primary analyses and wrote the manuscript with feedback and contributions from all co-authors. TJL supervised the study, served as AMPR Principal Investigator (PI), and assisted with refining the methods and interpreting results. CGA and TJL processed the AMPR data. SCvdH served as PI for AVAPS, while RAF and CAH served as co-PIs for HSRL2. OOS processed the APR-3 data. SCvdH, RAF, OOS, LDC, SAC, and JRM assisted with refining the methods and interpreting results. SWF and GAS processed the AVAPS data. ST served as APR-3 PI.

**Competing interests**

The authors declare that they have no conflict of interest.

**Acknowledgements**

We are grateful to Hal Maring for financial support throughout the CAMP$^2$Ex deployment and data analyses, and to Jeff Reid for managing the CAMP$^2$Ex mission. We would also like to thank Wojciech Grabowski and an anonymous reviewer for their insightful comments and suggestions that helped improve this manuscript.

**Financial support**

CGA acknowledges funding from NASA Marshall Space Flight Center through Cooperative Agreement 80MSFC22M0001 with The University of Alabama in Huntsville. CGA's research was further supported by an appointment to the NASA Postdoctoral Program at NASA Marshall Space Flight Center, administered by Oak Ridge Associated Universities under contract with NASA, through contract 80HQTR21CA005.

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
