# Peer review of "Observed impacts of aerosol concentration on maritime tropical convection within constrained environments using airborne radiometer, radar, lidar, and dropsondes"

_EGUsphere, 2024_

## Referee Comment (RC2)

Review of *Observed impacts of aerosol concentration on maritime tropical convection within constrained environments using airborne radiometer, radar, lidar, and dropsondes* by Amiot et al. (2024).

Amiot et al. investigate the correlations between a number of aerosol, convective and environmental parameters observed on research flights during the CAMP2Ex campaign. They hypothesize that an increase in values of aerosol parameters leads to an increase of values of convective parameters under similar environmental conditions. I believe that the foundation of this manuscript has merit and would be a good addition to the convective invigoration literature. Especially interesting is the analysis of a large array of environmental and aerosol parameters and how they are correlated with convective metrics. However, I have some major concerns about the methodological approach and the structure of the manuscript that must be addressed.

Given these concerns I cannot recommend this manuscript for publication at this time. I believe that addressing my concerns is likely to require a substantial amount of time exceeding that typical of a major revision. However, if the authors do address these concerns, and decide to resubmit, I would be happy to review a resubmission.

The introduction and methods sections were honestly quite confusing and did not serve as good introduction to the science at hand or provide enough information to build trust in the data. In its current state this manuscript is of little value to anyone not directly involved with CAMP2Ex. While the results should be reproducible with the descriptions in the manuscript, I am not sure whether the methodological approach provides robust results. It seems that some of these issues might be due this manuscript being mostly the same as Chapter 4 in Amiot (2023), with little changes to make it suitable for journal publication. I.e., things that might have been described in preceding chapters of Amiot (2023) are missing from this manuscript.

Below I list comments with some suggestions that the authors might want to consider to improve the manuscript.

**Major Comments**

1.  The introduction needs to be reworked. For instance, the three paragraphs from ll. 51-110 do not really belong in the introduction and do not introduce the reader to the general scientific topic. I suggest that the authors move this description to the methods section.

    Furthermore, the literature review is very brief. This study tries to 'contribute knowledge to long-standing questions of aerosol influences on convection' through

the analysis of observations. However, the referenced literature almost exclusively concerns modeling and theoretical work. I suggest that the authors include more observational studies in their introduction. Some examples are: Lin et al. (2006), Fan et al. (2018), Veals et al. (2022), and Zang et al. (2023).

2. Currently, the manuscript lacks any information about the CAMP2Ex campaign. It is not even mentioned when the campaign took place or what type of convection is investigated ('maritime tropical convection' is very broad)! The authors should not expect every reader to be familiar with this campaign. I suggest starting section 2 with a general description of the CAMP2Ex campaign.

   This description does not need to be overly long since the campaign has been described in detail in the referenced literature, but some basic description is needed. When and where did it take place? Some details about the P3, the instruments and the SFs. What were the meteorological conditions, are they comparable between the analyzed SFs? What types of clouds are we looking at? Why are SF 1-4 not included? These are all very basic questions that are needed to fully understand this manuscript but are not addressed at all.

3. A possible major concern with the data that needs to be addressed is the different scene length. Figure 1 suggests a bimodal distribution of scene lengths with 'short' scenes under 5 min long and longer scenes around 10 min long. Assuming that the P3 flies completely straight in each scene and at similar speed (impossible to know see comment 2), the longer scenes inevitably capture more meteorological variability than the short scenes. Any potential issues with differing scene times is not acknowledged until l. 452.

   This could lead to major sampling issues. For instance, the convective parameters could be representative of a cloud that is at one end of the scene, but the aerosol parameters could be significantly influenced by data at the other end of the scene. In such a case the scene's aerosol values do not represent the actual aerosol conditions at the location of the convection. Such issues become increasingly likely with longer scenes. With the authors providing no example data from a scene it is impossible to determine if this actually was an issue. I suggest that the authors investigate the sensitivity of their results to limiting the scene lengths to different times. For instance, for all scenes longer than 5 minutes, only the 5 minutes around the dropsonde are used (5 min serves as an example, I cannot reasonably say

whether this is a good threshold). Or the authors could set a maximum horizontal length for scenes appropriate for the convection being analyzed. Alternatively, the authors should demonstrate that different scene lengths do not impact the results. See the minor comment below on including a figure of a 'scene' that might help with this issue as well.

4. During CAMP2Ex a wide range of shallow to deep convection was observed. Aerosol impacts in these regimes might be vastly different. There is no description of a separation between these regimes. Are the stratified environmental parameters a good predictor of the type of convection? I understand that sample sizes are already small, however, I believe the authors should at least look at their results separated by different types of convection. Whether this should be included in the manuscript depends on the results.

5. ll. 152-153: This first hypothesis does not seem to be directly addressed anywhere. There is only some description based on the stratification in the scatter plots but at best this is a weak indicator of correlations. I think it would be beneficial if the authors test how the chosen environmental parameters correlate with the chosen convective (aerosol) parameters for their data set as was promised by this hypothesis.

6. Based on some of the comments that I have classified as minor below and my concerns about the methodological approach, I believe some of the conclusions are only weakly supported by the results. Some conclusions are very speculative based on just a few data points. This issue could be overcome by focusing on specific types of convection and using a more standardized definition of a scene. Currently, it seems that it is quite possible that outliers on which the authors base some conclusions are at least partially due to different scene length. Furthermore, the manuscript could benefit from being more focused on which specific environmental and aerosol parameters are actually good predictors of the convective intensity/frequency. The most we learn is that some convective parameters correlate with some aerosol parameters, for lapse rate and K-index stratifications. I suggest expanding these types of analysis.

7. Are the authors employing the sensitivity tests of their environmental binning in response to the concerns raised about environmental binning techniques in Varble et al. (2023)? The authors might want to make this connection since at least some of

the results appear to be robust across the sensitivity tests. (And those that are not highlight the importance of such sensitivity tests!)

**Minor comments**

8. In the short summary, the terms 'microwave-frequency' and 'K-index' are too technical for the more general audience. I suggest removing 'microwave-frequency' altogether and replacing 'K-index' with something like 'convective potential'.

9. l. 82: The authors acknowledge that convective intensity refers to peak updraft velocity. They should also acknowledge that the convective parameters they analyze (in particular reflectivity) can have changes unrelated to convective intensity as described in Varble et al. (2023).

10. Does Figure 1 show the length of all scenes during all flights? The number of occurrences adds up to significantly more than 144. Even counting the bars conservatively, there seem to be more than 300 scene lengths in this figure. Why do the authors include the length of scenes that are not analyzed? I suggest reducing the data in this figure to the 144 analyzed scenes. If the authors standardize their scenes this figure could be removed altogether.

11. I suggest a figure displaying a 'scene'. This could help addressing some of the major comments. There are a lot of measurements that could be shown in such a figure, and they cannot all reasonably be included. However, I suggest time-height series (curtains) of reflectivity of at least one APR-3 frequency (at nadir, see Fig. 7 in Reid et al., 2023), time-height series of HSRL2 backscatter for one wavelength, and some timeseries from Table 1. Maybe a dropsonde profile could be included as well (see Fig. 7 in Reid et al., 2023), with corresponding values for the environmental parameters.

12. Was any testing done whether dropsondes passed through clouds? Such dropsondes are unlikely to be representative of the environmental background and should be used with caution when determining convective parameters.

13. The authors acknowledge the inconsistency of 'modified CAPE' due to dropsondes launching at different altitudes, but do not seem to address this issue afterwards. Currently, it seems that CAPE would most strongly correlate with launch altitude

and not convective parameters. The authors could consider using (modified) normalized CAPE instead, i.e. normalize CAPE by the altitude of the P3/dropsonde launch.

14. ll. 187-188: 'would likely be less than true CAPE', remove 'likely', since it would always be less than true CAPE , if 'the P3 did not fly above the EL during any science flight'.

15. ll. 188-190: Without any description of the meteorology and cloud environment, it is impossible to understand the importance of this statement.

16. ll. 193,194,250 etc.: I personally would remove any mention of the Python packages from the main body of the manuscript. This might confuse some readers not familiar with Python (e.g., what is 'np.percentile'?). Instead, I suggest mentioning these packages and what they calculate in the data availability statement. The authors should then also mention what they use to calculate Pearson correlation coefficients and p-values.

17. Table 1: Please add units.

18. ll. 217-220: '[...] due to their direct association with peak convective intensity' please add a reference for this statement.

19. ll. 217: In my experience, maximum values in such observations can potentially represent significant outliers. I suggest testing the sensitivity of the results to high percentile values (e.g., $99^{th}$ and/or $95^{th}$) of these observations or demonstrating that maximum values do represent the actual environment well.

20. From my understanding the parameter $Pixels_{Ku}$ should strongly correlate with scene length because longer scenes would inevitably contain more such pixels. This is later acknowledged in l. 452. Based on this understanding of $Pixels_{Ku}$, the analysis of $Pixels_{Ku}$ appears to me to be largely meaningless. I suggest trying to normalize this parameter by scene length or standardizing scene length altogether as described above.

21. Figures 3, 6, and 9 subplots should be labeled with (a), (b) etc. and referred to as such in the text. Furthermore, I suggest increasing the label sizes in these figures

and adding gridlines. The gridlines should make it easier to see which data points are actually within the ranges described in the text.

22. Did the authors consider the measurement uncertainties when determining the correlation coefficients? Or are measurement uncertainties so small that they are of no concern? If not, the authors could use Monte Carlo type simulations to estimate the uncertainty of their correlation coefficients. I.e., simulate many random instances of the data set based on the uncertainties and determine a mean and variance of the correlation coefficient for all parameters.

23. Maybe this is a misunderstanding on my part: In Figure 3, I counted the number of data points. They do not match what is shown in the corresponding figures (Figure 2 for the top row). For instance, in Figure 2 it says there should be 16 data points for medium K-index when comparing maximum AMPR CLW vs. 532 nm AOT. However, in Figure 3 top right I only count 13 data points for medium K-index (green). What is the reason for this discrepancy? This seems to be happening in almost all scatter plots. Are some data points hidden below others (although that seems unlikely)? It does not appear that ll. 240-244 explain this.

24. l. 339: Figure 4 shows a low correlation coefficient and high p-value for medium LR850-500 and PCT19 vs. Bsc532. This statement seems to be incorrect. Figure 3 bottom right confirms that there is no correlation. Do the authors mean high instead of 'medium'?

25. l. 340: There are only six data points in the described area and one of those is associated with low values. I personally would not draw the subsequent conclusion from such few data points. If one randomly draws 6 data points from 37 low, 44 medium, and 40 high values (as shown in Figure 4, although there are fewer points in figure 3 bottom left, see comment 23), the chance of there being exactly one low data point in those 6 is about 30%! Thus, it is not too unlikely that the shown behavior happened by pure chance.

26. ll. 353-354: This sentence is incorrect because there is no correlation for the medium and high bins, i.e. there is no 'increase in PCT19 with increasing Bsc532'. The sentence afterwards describes better what is actually going on.

27. Figure 6 bottom row: the correlations for medium and high environmental parameters appear to be high because of single outlier values. The authors might want to consider adding the actual regression lines as shown in Chapter 4 of Amiot (2023).

28. ll. 439: Again, could this not just be because it is a longer scene, saying nothing about the actual convective frequency?

29. ll. 448-451: but there is also two data points with low-to-medium K-index and low backscatter but very high convective frequency. These data points do not support the conclusion in this sentence.

**Typographical:**

30. l. 111: 'have been'
31. l. 137: 'that was deployed on'
32. l. 157: I suggest removing 'radiometer-retrieved'
33. l. 171: QC (quality control?) has not been defined.
34. l. 214: 'Nine remote-sensing parameters related to convective intensity', I believe that this should be 'eight' instead of 'nine' as shown in table 1 and mentioned elsewhere in the text.
35. l. 292: 'increase'
36. l. 293: 'is associated with', 'in association with' or 'associated with' ?
37. ll. 292-293: something else seems wrong about this sentence, please correct.

**References**

- Amiot, C. G.: Airborne passive microwave geophysical retrievals and applications in assessing environmental and aerosol impacts on maritime convection. Ph.D. dissertation, Dept. of Atmospheric and Earth Science, The University of Alabama in Huntsville, Huntsville, AL, 176 pp, https://louis.uah.edu/uah-dissertations/278/, 2023
- Fan J, Rosenfeld D, Zhang Y, Giangrande SE, Li Z, Machado LAT, Martin ST, Yang Y, Wang J, Artaxo P, Barbosa HMJ, Braga RC, Comstock JM, Feng Z, Gao W, Gomes HB,

Mei F, Pöhlker C, Pöhlker ML, Pöschl U, de Souza RAF. Substantial convection and precipitation enhancements by ultrafine aerosol particles. Science. 2018 Jan 26;359(6374):411-418. doi: 10.1126/science.aan8461

- Lin, J. C., T. Matsui, R. A. Pielke Sr., and C. Kummerow (2006), Effects of biomass-burning-derived aerosols on precipitation and clouds in the Amazon Basin: a satellite-based empirical study, *J. Geophys. Res.*, 111, D19204, doi:10.1029/2005JD006884

- Reid, J. S., and Coauthors: The coupling between tropical meteorology, aerosol lifecycle, convection and the energy budget: The Cloud, Aerosol and Monsoon Processes Philippines Experiment (CAMP2Ex). B. Am. Meteorol. Soc., 106, E1179–E1205, https://doi.org/10.1175/BAMS-D-21-0285.1, 2023

- Varble, A. C., Igel, A. L., Morrison, H., Grabowski, W. W., and Lebo, Z. J.: Opinion: A critical evaluation of the evidence for aerosol invigoration of deep convection, Atmos. Chem. Phys., 23, 13791–13808, https://doi.org/10.5194/acp-23-13791-2023, 2023

- Veals, P. G., A. C. Varble, J. O. H. Russell, J. C. Hardin, and E. J. Zipser, 2022: Indications of a Decrease in the Depth of Deep Convective Cores with Increasing Aerosol Concentration during the CACTI Campaign. J. Atmos. Sci., 79, 705–722, https://doi.org/10.1175/JAS-D-21-0119.1

- Zang, L., Rosenfeld, D., Pan, Z., Mao, F., Zhu, Y., Lu, X., & Gong, W. (2023). Observing aerosol primary convective invigoration and its meteorological feedback. *Geophysical Research Letters*, 50, e2023GL104151. https://doi.org/10.1029/2023GL104151

---

## Author Comment (AC1)

**Responses to Reviewers**

We would like to thank the Reviewers for taking the time to read our manuscript and provide thorough feedback to help improve our study and the presentation of the results. Using your valuable feedback, we have made several substantial changes to our study to mitigate some sources of uncertainty, and we have updated the manuscript text to better convey the introduction and results of our study. In particular, we have standardized the duration of a given "scene" to be 10 minutes around a given dropsonde. Following this change, we removed the histogram of scene durations previously presented as Fig. 1 and have added a new Fig. 1 which shows relevant data from a single example scene. The Purpose and Background (section 1) has been revised to give a more-thorough overview of the prior literature and the motivation for our study, with information about specific environmental parameters moved to section 2. We now also utilize the $95^{th}$ percentile, rather than the maximum, of each AMPR and APR-3 convective parameter in a given scene when comparing with the mean aerosol parameters to avoid potential influences from outlier values. We've also normalized all (modified) CAPE values by dropsonde altitude and presented this new variable as $NCAPE_{mod}$. This inclusion of the $95^{th}$ percentile threshold, $NCAPE_{mod}$, and the standardized scene durations have significantly modified our results, which we present throughout sections 3–5.

Our responses to specific Reviewer comments can be found below. In the following discussion, Reviewer comments are presented in italicized font and our response immediately follows in standard font.

**Reviewer #1:**

*Review of "Observed impacts of aerosol concentration on maritime tropical convection within constrained environments using airborne radiometer, radar, lidar, and dropsondes" by Amiot et al.*

*Recommendation: accept after revisions*

*Overall evaluation: This paper investigates links between various observed atmospheric parameters and the moist convection strength with the overall goal to understand aspects of the so-called convection invigoration. In other words, the motivation is to explore links between factors that in theory can affect moist convection and observed convection strength. The observations come from the CAMP2Ex experiment.*

*First, I have to say that I am not the right person to review this submission. Although I was involved over the last decade in the discussions of the invigoration conundrum (and for that reason I am signing my review), I feel someone with more expertise in atmospheric observations should also be involved. In particular, I feel the observations lack estimates of their uncertainty. Since this is not my area, I am not sure what to suggest. One possible suggestion is to use a spread of the observations near (in space and time) of the convective event. But this aspect can only address the sampling problem, that is, an uncertainty of a single observation. Another important aspect is an accuracy of the observation itself or uncertainty of a retrieval algorithm. I am not capable to assess if that aspect is appropriately addressed in the submission.*

*Below I discussed two major points concerning this submission, and follow with several specific comments that require authors' attention.*

We sincerely thank you for taking the time to review our manuscript and provide helpful feedback. We have included more information about the uncertainties associated with each instrument utilized in our study on Lines 153–161.

*Specific major comments.*

*I have two main issues concerning the motivation and interpretation of results. For the motivation, the convective invigoration discussed in the introduction is poorly explained. There are important published studies that discuss and criticize the original convective invigoration proposal of Andreae et al. (2004) and Resenfeld et al. (2008), including my own papers, that should be included in the introduction. The recent review paper by Varble et al. (ACP 2023, https://doi.org/10.5194/acp-23-13791-2023) should definitely be cited for that.*

We have refined our literature review of convective invigoration from increased aerosol concentrations throughout section 1, including an additional paragraph starting on Line 82 to discuss studies that have presented counterarguments for this convective invigoration, especially

above the environmental freezing level. This updated paragraph includes references to Grabowski and Morrison, (2016, 2017, 2020) and Varble et al., (2023).

*For the interpretation, I strongly object the suggestion in the summary section 5 that the results support the notion of enhanced aerosol concentrations may invigorate convection (e.g., lines 501 and 510). The key problem is that the correlation does not imply causality. Is it possible that higher aerosol concentrations simply occur in conditions supporting stronger convection (e.g., higher CAPE)? The text in lines 512-515 seems to suggest that such a conclusion might be valid. One suggestion would be to explain the correlation versus causality conundrum in the introduction, and then try to use the results to shed some light on the problem.*

We have added a sentence to section 1 (Lines 134–136) to emphasize that correlation does not imply causality, and we now refer back to this point throughout the manuscript. We have also pulled back somewhat from asserting such conclusions in the summary section, and we now briefly mention these potential implications but focus more on the correlations as indicators of potentially interesting trends that are worth examining in future work, rather than providing definitive conclusions on their own. We have highlighted some of the most-interesting trends in the scatterplots more clearly in the updated summary as well. You are correct that, despite our environmental stratifications, there is still a chance that there is a convective dependence on the values within each group; e.g., "low high" CAPE versus "high high" CAPE for the high CAPE group. We acknowledge that correlation does not indicate causality as noted in the manuscript, and this is meant to inspire additional work rather than be a conclusive/final study.

*However, I have to admit that the discussion of the results in sections 3 and 4 are difficult for me to follow. Specifically, I do not see any trends in figs. 3, 6 and 9, just scattered data points. Does that suggest that the overall outcome of the study is inconclusive? Is there a better way to present the results? See 7 below.*

We have included additional emphasis in our updated manuscript that correlation does not indicate causation, and thus we aren't able to make definitive conclusions about aerosol impacts on convection solely based on our results. However, the conclusions we present in the updated manuscript may inform future studies about potentially interesting relationships between a wide range of convective and aerosol parameters that may be worthy of further investigation. There were indeed several comparisons that yielded little to no correlation, but others provided statistically significant correlations (though, many were hindered by a relatively small sample size, and some trends did not stand out very clearly, as you note). Please see our response to your comment #7 below regarding the presentation of the results.

*Specific comments.*

*1.     The paragraph starting in l. 68 and the reference to Mulholland et al (2021) in particular. The argument here is wrong as discussed in Grabowski (QJRMS 2023), see the summary Fig. 17 there in particular. Please revise keeping in mind that the argument in my paper applies to convective boundary layer over land. Such an argument might not always apply to oceanic convection with weak surface buoyancy forcing. The above comment is also relevant to the discussion around l. 156. Specifically, it is unclear to me why higher LCL may favor stronger convection.*

We have modified our discussion in this paragraph (now starting on Line 115) to note that the conclusions of Mulholland et al., (2021) have been debated in subsequent literature (specifically citing Grabowski, 2023). We still examine LCL altitude in this paper to see what correlations result between the convective and aerosol parameters when binning the environments according to their LCL altitude (now with additional acknowledgements of the shortcomings associated with environmental binning, as discussed in Varble et al., 2023), but we note on Lines 131–132. that the exact impact of LCL altitude on convective intensity is still up for debate.

*2.     L. 156: It is unclear what you mean by "frequency" here and in other places (e.g., l. 214). Do you mean higher cloud cover? Please explain or remove to focus on the convective intensity alone.*

You are correct; "frequency" was meant to refer to the relative areal coverage of convective storms during a given flight scene. We have modified this wording to "prevalence" throughout the manuscript to avoid confusion, especially since we use "frequency" for other contexts (e.g., e.g., when discussing microwave frequencies).

*3.     L. 109 and later in the text. I was confused by units of CLW and I though it should be kg m$^{-3}$ (i.e., content). That was until I realized that this is the CLW path. I suggest to use CLWP throughout the manuscript to avoid confusion.*

We have retained the use of CLW to refer to integrated cloud liquid water path in the updated manuscript to maintain consistency with past AMPR-related literature, but we acknowledge the differences between CLW and CLWP and have now fully spelled out our definition of CLW on Line 120 as integrated cloud liquid water path.

*4.     The discussion in paragraph starting in l. 119. Please see me major comment above. Perhaps referring here to postulated "warm" and "cold" invigorations would be appropriate here. However, see the discussion in section 2 in Grabowski and Morrison (JAS 2020, p. 2567) and the review of Varble et al. (2023) already mentioned above.*

We have now included a discussion of the differences between convective invigoration from enhanced aerosol concentrations in warm-phase and cold-phase regions in the new paragraph that begins on Line 82.

*5.        L. 186 versus l. 63. Overall, CAPE refers to the buoyancy integral from the level of free convection to the equilibrium level as given by (1). Integrating buoyancy to the aircraft altitude as in (4) gives only a fraction of CAPE as pointed out in the submission. I feel the authors may want to refer to what I call "cumulative CAPE" (cCAPE), that is, how CAPE builds up in an adiabatic parcel as the parcel rises through the atmosphere. I feel this is a useful concept as shown, for instance, in Thomas et al. (ACP 2018, Fig. 4 in particular) and in some of my papers concerning convective dynamics (e.g., Grabowski and Morrison, ACP 2021, see Fig. 3 there). Clearly cCAPE depends on the aircraft altitude, so it is unclear how important that aspect is for the analysis. Can some large-scale analysis be used to extend calculations above the aircraft altitude to get the total CAPE?*

We have reevaluated our computation of "CAPE" in the revised manuscript given the concerns raised by all Reviewers.  However, rather than focusing on modified CAPE or cumulative CAPE, though the cumulative CAPE is certainly an interesting idea, we have normalized (modified) CAPE by the dropsonde launch altitude, similar to the methods of Blanchard, (1998).  This parameter, now labeled as $NCAPE_{mod}$ in the manuscript, is computed as

$$NCAPE_{mod} \ (m \ s^{-2}) = \frac{CAPE_{mod}}{z},$$

where z is the dropsonde launch altitude and

$$CAPE_{mod} \left( J \ kg^{-1} \right) = g \int_{z_{lfc}}^{z_{P3}} \frac{(T_v - T_{v,0})}{T_{v,0}} \ dz,$$

which was originally used as the variable "CAPE" in the previous version of the manuscript, with the "mod" subscript added to distinguish it from true CAPE.  The use of $NCAPE_{mod}$ allows for the vertical acceleration of the parcel to be evaluated directly, as discussed in Blanchard, (1998), and helps mitigate some of the influence of the varying dropsonde launch (i.e., P-3) altitude.  This information has been added to the manuscript in section 2, and the updated results are presented throughout sections 3 and 4

*6.        Table 1 should include units for all symbols.*

Added as suggested.

*7.        Figs. 5. 7, and 8. These are not figures, these are tables. It is difficult for me to draw any conclusions looking at them. Can they be shown as bar diagrams? I think the authors need to think about a better way to show those key results.*

We refer to these as figures since: 1) they utilize color shading from a colorbar throughout, and 2) they have multiple horizontal lines to create the grid structure, both of which are specifically mentioned in ACP's guidelines to be avoided for final published tables (https://www.atmospheric-chemistry-and-physics.net/submission.html#figurestables).  Since the color shading and grid pattern are essential for Figs. 2, 4, 5, 7, and 8, we classify them as figures.

In regard to the key results, we've considered several possible methods for presenting the results as concisely as possible, especially since there is a lot of information to include.  For example, in Fig. 2, we include every comparison of AMPR CLW with the aerosol conditions across all stratified environments in this study, along with Pearson correlation coefficients, p-values, and sample size.  We feel each of these is important to present to the reader to convey the full story of the study.  Expanding to other variables and sensitivity tests creates even more of the colored figures (e.g., as in the supplemental material).  We have attempted to highlight the key results in these figures using bolded text for the highest Pearson correlation coefficient magnitudes and strongest color shading for the most-significant p-values with the hope that it these would draw the reader's attention to the key relationships between the variables.  To highlight some of the key results further, we employ the scatterplots and draw connections between the color figures and the scatterplots as much as possible.  Utilizing other plot types (e.g., bar diagrams) without Figs. 2, 4, 5, 7, and 8 would limit the presentation of the full results.

**Reviewer #2**:

*Review of Observed impacts of aerosol concentration on maritime tropical convection within constrained environments using airborne radiometer, radar, lidar, and dropsondes by Amiot et al. (2024).*

*Amiot et al. investigate the correlations between a number of aerosol, convective and environmental parameters observed on research flights during the CAMP2Ex campaign. They hypothesize that an increase in values of aerosol parameters leads to an increase of values of convective parameters under similar environmental conditions. I believe that the foundation of this manuscript has merit and would be a good addition to the convective invigoration literature. Especially interesting is the analysis of a large array of environmental and aerosol parameters and how they are correlated with convective metrics. However, I have some major concerns about the methodological approach and the structure of the manuscript that must be addressed. Given these concerns I cannot recommend this manuscript for publication at this time. I believe that addressing my concerns is likely to require a substantial amount of time exceeding that typical of a major revision. However, if the authors do address these concerns, and decide to resubmit, I would be happy to review a resubmission.*

*The introduction and methods sections were honestly quite confusing and did not serve as good introduction to the science at hand or provide enough information to build trust in the data. In its current state this manuscript is of little value to anyone not directly involved with CAMP2Ex. While the results should be reproducible with the descriptions in the manuscript, I am not sure whether the methodological approach provides robust results. It seems that some of these issues might be due this manuscript being mostly the same as Chapter 4 in Amiot (2023), with little changes to make it suitable for journal publication. I.e., things that might have been described in preceding chapters of Amiot (2023) are missing from this manuscript.*

*Below I list comments with some suggestions that the authors might want to consider to improve the manuscript.*

We sincerely thank you for taking the time to review our manuscript and provide helpful feedback.

**Major Comments**

*1.      The introduction needs to be reworked. For instance, the three paragraphs from ll. 51-110 do not really belong in the introduction and do not introduce the reader to the general scientific topic. I suggest that the authors move this description to the methods section.*

We have moved these paragraphs to the Data and Methods section as suggested, and we have included additional information throughout the Purpose and Background section to bolster it.

*Furthermore, the literature review is very brief. This study tries to 'contribute knowledge to long-standing questions of aerosol influences on convection' through concerns modeling and theoretical work. I suggest that the authors include more observational studies in their introduction. Some examples are: Lin et al. (2006), Fan et al. (2018), Veals et al. (2022), and Zang et al. (2023).*

We have added these suggested references to the manuscript and referred to them in a new paragraph that we have added to the Purpose and Background section starting on Line 76.

*2.      Currently, the manuscript lacks any information about the CAMP2Ex campaign. It is not even mentioned when the campaign took place or what type of convection is investigated ('maritime tropical convection' is very broad)! The authors should not expect every reader to be familiar with this campaign. I suggest starting section 2 with a general description of the CAMP2Ex campaign. This description does not need to be overly long since the campaign has been described in detail in the referenced literature, but some basic description is needed. When and where did it take place? Some details about the P3, the instruments and the SFs. What were the meteorological conditions, are they comparable between the analyzed SFs? What types of clouds are we looking at? Why are SF 1-4 not included? These are all very basic questions that are needed to fully understand this manuscript but are not addressed at all.*

These omissions were definitely in error and occurred during the transition of Chapter 4 in Amiot (2023) to this manuscript, as you noted in one of your previous paragraphs.  We have included additional details about the CAMP$^2$Ex campaign on Lines 44–52, including descriptions of when the campaign took place, the types of clouds investigated, and the instruments that participated in the campaign.  Details regarding the exclusion of SFs 1–4 have been added on Lines 256–258.

*3.      A possible major concern with the data that needs to be addressed is the different scene length. Figure 1 suggests a bimodal distribution of scene lengths with 'short' scenes under 5 min long and longer scenes around 10 min long. Assuming that the P3 flies completely straight in each scene and at similar speed (impossible to know see comment 2), the longer scenes inevitably capture more meteorological variability than the short scenes. Any potential issues with differing scene times is not acknowledged until l. 452. This could lead to major sampling issues. For instance, the convective parameters could be representative of a cloud that is at one end of the scene, but the aerosol parameters could be significantly influenced by data at the other end of the scene. In such a case the scene's aerosol values do not represent the actual aerosol conditions at the location of the convection. Such issues become increasingly likely with longer scenes. With the authors providing no example data from a scene it is impossible to determine if this actually was an issue. I suggest that the authors investigate the sensitivity of their results to limiting the scene lengths to different times. For instance, for all scenes longer than 5 minutes, only the 5 minutes around the dropsonde are used (5 min serves as an example, I*

*cannot reasonably say whether this is a good threshold). Or the authors could set a maximum horizontal length for scenes appropriate for the convection being analyzed. Alternatively, the authors should demonstrate that different scene lengths do not impact the results. See the minor comment below on including a figure of a 'scene' that might help with this issue as well.*

We fully agree about the benefits of using standardized scene durations and acknowledge the shortcomings of using varying scene durations. As a result, we have now standardized all scene durations to be 10 minutes long, ±5 minutes from the associated dropsonde launch time. This allows time for ample observations around the dropsonde location and gives time for the dropsonde to descend (most CAMP$^2$Ex dropsonde descent durations were 5–10 minutes; Vömel et al., 2020). Since these new standardized scene durations involve the combination of multiple APR-3 scan files in many instances, there were some cases where data were unavailable at the start and/or end of a given scene (e.g., P-3 was in a turn). To accommodate some of these cases where reliable data became available shortly before and/or after the ±5-minute window, we have allowed up to 1-minute (i.e., 10%) grace in the scene duration. That is, any scene durations < 9 minutes or > 11 minutes were discarded from the analysis, which amounted to 47 of the 144 dropsondes previously considered. We have included a description of this uncertainty on Lines 259–263. When combined with five additional dropsondes excluded due to potential cloud contamination (please see our response on page 12), our new dropsonde sample size is 92.

Your comment about the possibility for convective metrics to be influenced by one end of the scene while aerosol metrics are influenced by the other end of the scene are possible, and we have not specifically examined spatial offsets between the convective and aerosol metrics in our scenes. However, we attempt to mitigate these effects through the use of a scene-averaged mean value for each aerosol parameter, rather than relying on maximum values, and we now use the 95$^{th}$ percentile of the convective metrics, following your recommendation in a different comment. As you've correctly stated, the potential impact of these spatial offsets would increase with increasing scene duration, and our new standardization of the scene durations helps remove the longest scenes, where were previously ~18 minutes long (albeit, while also increasing the duration of shorter scenes that were previously ~2 minutes long, for which the effects of spatial offsets were relatively limited).

*4. During CAMP2Ex a wide range of shallow to deep convection was observed. Aerosol impacts in these regimes might be vastly different. There is no description of a separation between these regimes. Are the stratified environmental parameters a good predictor of the type of convection? I understand that sample sizes are already small, however, I believe the authors should at least look at their results separated by different types of convection. Whether this should be included in the manuscript depends on the results.*

We agree that such an analysis would be beneficial, but we have kept our focus on the campaign-wide statistics for the time being, largely due to the limited sample size as you note (which was further reduced from 144 scenes to 92 scenes in our updated manuscript). However, we have noted the different types of clouds observed during CAMP$^2$Ex on Lines 47–48 and added this separation as future work on Lines 595–596.

*5.       ll. 152-153: This first hypothesis does not seem to be directly addressed anywhere. There is only some description based on the stratification in the scatter plots but at best this is a weak indicator of correlations. I think it would be beneficial if the authors test how the chosen environmental parameters correlate with the chosen convective (aerosol) parameters for their data set as was promised by this hypothesis.*

We agree that this type of analysis would be interesting and beneficial, but our focus in this study was on the convective-aerosol correlations within the stratified environmental groups, along with some (secondary) discussion of the observed increase or decrease in convective intensity/prevalence associated with the high, medium, and low groups for each environmental parameter in the scatterplots.  We have updated the wording around Lines 127–128 and 133–134 in the updated manuscript to convey this more clearly.

*6.       Based on some of the comments that I have classified as minor below and my concerns about the methodological approach, I believe some of the conclusions are only weakly supported by the results. Some conclusions are very speculative based on just a few data points. This issue could be overcome by focusing on specific types of convection and using a more standardized definition of a scene. Currently, it seems that it is quite possible that outliers on which the authors base some conclusions are at least partially due to different scene length. Furthermore, the manuscript could benefit from being more focused on which specific environmental and aerosol parameters are actually good predictors of the convective intensity/frequency. The most we learn is that some convective parameters correlate with some aerosol parameters, for lapse rate and K-index stratifications. I suggest expanding these types of analysis.*

In our updated manuscript, we have placed less emphasis on the outlier values, instead focusing on more-general trends and discussing the influences outliers may have on some of the results, especially those associated with a relatively small sample size.  While we have standardized the scene durations in the updated manuscript, which helps to mitigate some of the uncertainty, we have still shifted the focus away from the outlier values compared to the previous version of the manuscript.  Our main goal was to begin with this investigation and all results of our analyses are included for the interested reader.  However, we agree that expanded analyses of key environmental parameters would be beneficial, and we have listed this as an avenue for future work.

*7.       Are the authors employing the sensitivity tests of their environmental binning in response to the concerns raised about environmental binning techniques in Varble et al. (2023)? The authors might want to make this connection since at least some of the results appear to be robust across the sensitivity tests. (And those that are not highlight the importance of such sensitivity tests!)*

The initial idea for performing the sensitivity tests was developed in 2022 [i.e., prior to the publication of Varble et al., (2023)], but we agree that the ideas raised in Varble et al., (2023)

should be referenced in our manuscript and the results herein should be connected to their study. We have added references to Varble et al., (2023) throughout the updated manuscript.

*Minor comments*

*8.      In the short summary, the terms 'microwave-frequency' and 'K-index' are too technical for the more general audience. I suggest removing 'microwave-frequency' altogether and replacing 'K-index' with something like 'convective potential'.*

Modified as suggested.

*9.      l. 82: The authors acknowledge that convective intensity refers to peak updraft velocity. They should also acknowledge that the convective parameters they analyze (in particular reflectivity) can have changes unrelated to convective intensity as described in Varble et al. (2023).*

This is an important point to consider. We have added a note on Lines 136–138 about how the radar- and radiometer-based convective metrics may vary due to factors not specifically tied to peak updraft intensity.

*10.      Does Figure 1 show the length of all scenes during all flights? The number of occurrences adds up to significantly more than 144. Even counting the bars conservatively, there seem to be more than 300 scene lengths in this figure. Why do the authors include the length of scenes that are not analyzed? I suggest reducing the data in this figure to the 144 analyzed scenes. If the authors standardize their scenes this figure could be removed altogether.*

This was the result of a coding typo in our previous draft, but we have removed this figure from the updated manuscript due to the standardization of scene times in the new results, following your recommendation.

*11.      I suggest a figure displaying a 'scene'. This could help addressing some of the major comments. There are a lot of measurements that could be shown in such a figure, and they cannot all reasonably be included. However, I suggest time-height series (curtains) of reflectivity of at least one APR-3 frequency (at nadir, see Fig. 7 in Reid et al., 2023), time-height series of HSRL2 backscatter for one wavelength, and some timeseries from Table 1. Maybe a dropsonde profile could be included as well (see Fig. 7 in Reid et al., 2023), with corresponding values for the environmental parameters.*

We have developed a new figure illustrating a single 10-minute "scene," which is presented as the new Fig. 1 in the manuscript. Since our study focuses on APR-3 composite $Z_H$ throughout a given scene, we have presented Ku-band composite $Z_H$ in Fig. 1 rather than a time-height series of reflectivity. Likewise, we have included AMPR CLW values from each AMPR pixel throughout the scene. Following your suggestion, we included the corresponding dropsonde profile and time-height series of HSRL2 532-nm backscatter for this scene.

*12. Was any testing done whether dropsondes passed through clouds? Such dropsondes are unlikely to be representative of the environmental background and should be used with caution when determining convective parameters.*

We had not screened dropsondes for the presence of clouds in our prior submission, but we have done so for the updated manuscript. Since the uncertainty in a given AVAPS relative humidity (RH) measurement is approximately 3% (Freeman et al., 2020), we examined the presence of RH values > 97% in each profile. While some dropsondes did indeed pass through clouds, most of them were relatively brief (e.g., < 5% of the total sounding). However, we identified five dropsondes where more than 20% of the dropsonde profile occurred in-cloud, and we have removed these from our updated analyses to avoid their potential contamination of the results, as you note. We have added a sentence on Lines 151–154 explaining this. When combined with 47 additional dropsondes excluded due to limitations with defining a 10-minute scene around them (please see our response on page 9), our new dropsonde sample size is 92.

*13. The authors acknowledge the inconsistency of 'modified CAPE' due to dropsondes launching at different altitudes, but do not seem to address this issue afterwards. Currently, it seems that CAPE would most strongly correlate with launch altitude and not convective parameters. The authors could consider using (modified) normalized CAPE instead, i.e. normalize CAPE by the altitude of the P3/dropsonde launch.*

We have reevaluated our computation of "CAPE" in the revised manuscript given the concerns raised by all Reviewers. Following your suggestion, we have normalized (modified) CAPE by the dropsonde launch altitude, similar to the methods of Blanchard, (1998). This parameter, now labeled as $NCAPE_{mod}$ in the manuscript, is computed as

$$NCAPE_{mod} \ (m \ s^{-2}) = \frac{CAPE_{mod}}{z},$$

where z is the dropsonde launch altitude and

$$CAPE_{mod} \ (J \ kg^{-1}) = g \int_{z_{lfc}}^{z_{P3}} \frac{(T_v - T_{v,0})}{T_{v,0}} \ dz,$$

which was originally used as the variable "CAPE" in the previous version of the manuscript, with the "mod" subscript added to distinguish it from true CAPE. The use of $NCAPE_{mod}$ allows

for the vertical acceleration of the parcel to be evaluated directly, as discussed in Blanchard, (1998), and helps mitigate some of the influence of the varying dropsonde launch (i.e., P-3) altitude. This information has been added to the manuscript in section 2, and the updated results are presented throughout sections 3 and 4.

*14.    ll. 187-188: 'would likely be less than true CAPE', remove 'likely', since it would always be less than true CAPE , if 'the P3 did not fly above the EL during any science flight'.*

Modified as suggested.

*15.    ll. 188-190: Without any description of the meteorology and cloud environment, it is impossible to understand the importance of this statement.*

We have added additional descriptions of the cloud environments on Lines 47–48.

*16.    ll. 193,194,250 etc.: I personally would remove any mention of the Python packages from the main body of the manuscript. This might confuse some readers not familiar with Python (e.g., what is 'np.percentile'?). Instead, I suggest mentioning these packages and what they calculate in the data availability statement. The authors should then also mention what they use to calculate Pearson correlation coefficients and p-values.*

The references to specific Python packages were moved to the data availability statement as suggested, and details were added therein regarding the calculations of Pearson correlation coefficients and the associated p-values.

*17.    Table 1: Please add units.*

Added as suggested

*18.    ll. 217-220: '[…] due to their direct association with peak convective intensity' please add a reference for this statement.*

We have added a reference to Kollias et al., (2001) on Line 273.

*19.      ll. 217: In my experience, maximum values in such observations can potentially represent significant outliers. I suggest testing the sensitivity of the results to high percentile values (e.g., 99th and/or 95th) of these observations or demonstrating that maximum values do represent the actual environment well.*

We have updated our analysis to use the 95$^{th}$ percentile of the AMPR and APR-3 convective metrics from a given scene, rather than using the maximum value.

*20.      From my understanding the parameter PixelsKu should strongly correlate with scene length because longer scenes would inevitably contain more such pixels. This is later acknowledged in l. 452. Based on this understanding of PixelsKu, the analysis of PixelsKu appears to me to be largely meaningless. I suggest trying to normalize this parameter by scene length or standardizing scene length altogether as described above.*

You are correct about the direct impact scene duration would have on Pixels$_{Ku}$.  We have mitigated this effect with the new standardized scene durations of 10 minutes each (please see our response on page 9).

*21.      Figures 3, 6, and 9 subplots should be labeled with (a), (b) etc. and referred to as such in the text. Furthermore, I suggest increasing the label sizes in these figures and adding gridlines. The gridlines should make it easier to see which data points are actually within the ranges described in the text.*

All modified as suggested.

*22.      Did the authors consider the measurement uncertainties when determining the correlation coefficients? Or are measurement uncertainties so small that they are of no concern? If not, the authors could use Monte Carlo type simulations to estimate the uncertainty of their correlation coefficients. I.e., simulate many random instances of the data set based on the uncertainties and determine a mean and variance of the correlation coefficient for all parameters.*

The measurement and/or retrieval uncertainties in the AMPR, APR-3, and HSRL2 (i.e., the instruments considered in the correlation coefficient calculations) data were deemed negligible for the purposes of this study.  We have listed the uncertainty values for each of these instruments, along with AVAPS, on Line 155–161 in the updated manuscript and noted how they are relatively small.

*23.      Maybe this is a misunderstanding on my part: In Figure 3, I counted the number of data points. They do not match what is shown in the corresponding figures (Figure 2 for the top row). For instance, in Figure 2 it says there should be 16 data points for medium K-index when comparing maximum AMPR CLW vs. 532 nm AOT. However, in Figure 3 top right I only count 13 data points for medium K-index (green). What is the reason for this discrepancy? This seems to be happening in almost all scatter plots. Are some data points hidden below others (although that seems unlikely)? It does not appear that ll. 240-244 explain this.*

This may have been caused by an additional level of NaN-data masking that was present in the scatterplots but not the correlation figures.  Based on a visual check, this discrepancy is not present in the updated manuscript draft (i.e., the parenthesized values in the correlation figures match the number of data points in the corresponding scatterplots).

*24.      l. 339: Figure 4 shows a low correlation coefficient and high p-value for medium LR850-500 and PCT19 vs. Bsc532. This statement seems to be incorrect. Figure 3 bottom right confirms that there is no correlation. Do the authors mean high instead of 'medium'?*

Yes, thank you for catching this.  The previous statement was mistakenly based on $Bsc_{355}$, rather than $Bsc_{532}$, versus $PCT_{19}$ when binned by $LR_{850-500}$.  However, this statement is no longer present in the updated discussion.

*25.      l. 340: There are only six data points in the described area and one of those is associated with low values. I personally would not draw the subsequent conclusion from such few data points. If one randomly draws 6 data points from 37 low, 44 medium, and 40 high values (as shown in Figure 4, although there are fewer points in figure 3 bottom left, see comment 23), the chance of there being exactly one low data point in those 6 is about 30%! Thus, it is not too unlikely that the shown behavior happened by pure chance.*

The phrasing "$Bsc_{532} > 2$ $Mm^{-1}$ $sr^{-1}$ or $PCT_{19} > 240$ K" was meant to refer to all data points that met either condition, not necessarily both.  That is, 39 data points were within the described region of the plot, which did indeed differ from Fig. 4 as you noted and as discussed in our response to comment 23.  Of those 39 data points, 32 of them were associated with medium or high $LR_{850-700}$, which was the reason behind our original statement that a "vast majority" of the data points were associated with medium or high $LR_{850-700}$.  However, as discussed in our response to comment 23, we have clarified the discrepancy between the values in Figs. 3 and 4 (along with all other similar figure pairs within the manuscript) and all results have been updated.

*26.    ll. 353-354: This sentence is incorrect because there is no correlation for the medium and high bins, i.e. there is no 'increase in PCT19 with increasing Bsc532'. The sentence afterwards describes better what is actually going on.*

This statement is no longer present in the updated discussion.

*27.    Figure 6 bottom row: the correlations for medium and high environmental parameters appear to be high because of single outlier values. The authors might want to consider adding the actual regression lines as shown in Chapter 4 of Amiot (2023).*

We have added these regression lines to Figs. 3, 6, and 9 and have updated their associated discussions.

*28.    ll. 439: Again, could this not just be because it is a longer scene, saying nothing about the actual convective frequency?*

You are correct.  The frequency of convection could be directly related to scene duration, which is a main reason why we now standardize scene duration (please see our response on page 9).

*29.    ll. 448-451: but there is also two data points with low-to-medium K-index and low backscatter but very high convective frequency. These data points do not support the conclusion in this sentence.*

This statement is no longer present in the updated discussion.

***Typographical:***

*30.    l. 111: 'have been'*
*31.    l. 137: 'that was deployed on'*
*32.    l. 157: I suggest removing 'radiometer-retrieved'*
*33.    l. 171: QC (quality control?) has not been defined.*
*34.    l. 214: 'Nine remote-sensing parameters related to convective intensity', I believe that this should be 'eight' instead of 'nine' as shown in table 1 and mentioned elsewhere in the text.*
*35.    l. 292: 'increase'*
*36.    l. 293: 'is associated with', 'in association with' or 'associated with'?*
*37.    ll. 292-293: something else seems wrong about this sentence, please correct.*

All modified as suggested.  Thank you for catching these!

---

## Referee Report (RR1)

Review of revised manuscript "Observed impacts of aerosol concentration on maritime tropical convection within constrained environments using airborne radiometer, radar, lidar, and dropsondes" by Amiot et al.

Recommendation: accept after revisions

Overall evaluation: I think the paper has improved. Some of my comments on the observational uncertainty and interpretation of the observations (e.g., correlation versus causation) are now addressed (at least as far as I can tell). However, I claim that the introduction and motivation for the study presents an unclear (or even misleading) picture of the invigoration conundrum. In my opinion, the introduction is written from the perspective of a person who believes that the convective invigoration in polluted environments (all other factor being equal) is a proven effect. On physical grounds, the pollution-induced invigoration has little merit, especially the so-called cold invigoration that strongly depends on details of the freezing and condensate off-loading aloft (see section 2 in Grabowski and Morrison *JAS* 2020 and discussions in Igel and van den Heever *GRL* 2021 and Varble et al. *ACP* 2023). Below I provide several specific comments that the authors should address before the manuscript is accepted.

Specific comments on the introduction.

The introduction of the pollution-induced invigoration in the third paragraph of the opening section should start with two seminal papers that initiated the discussion: Andreae et al. (Science 2004, not listed in the manuscript) and Rosenfeld et al. (Science 2008). The brief review of studies trying to prove and disprove the impact in observations and modeling should follow. However, the physical basis of the invigoration should be also brought into the picture following the discussion in the papers listed above. It would be appropriate to point out that the warm-phase invigoration depends on the finite supersaturation within cloud updrafts (because reducing supersaturation in polluted clouds increases buoyancy), with the condensation rate depending only on the updraft velocity as long as the supersaturation is equal to the quasi-equilibrium supersaturation (see section 2b in Grabowski and Morrison *JAS* 2020). The cold-phase invigoration critically depends on the details of the frozen precipitation off-loading aloft because the latent heating due to freezing approximately balances the weight of the liquid water carried across the melting level (see section 2a in Grabowski and Morrison as well as Igel and van den Heever and Varble et al. papers). I feel a correct motivation for the study under review is important for providing a proper context. It also reflects in my view rather mixed results discussed in the paper, perhaps because details of the physical mechanisms involved (finite supersaturations, freezing and off-loading cloud condensate aloft) are practically impossible to document in observations. However, I feel the study is an important contribution to the problem, but a proper perspective of the past research is important.

Specific detailed comments.

1. L. 44: What do you mean by "environmental contexts". Please explain.

2. L. 65. The warm-phase invigoration is not defined. L. 91: the same for cold-phase invigoration.

3. When you bring Fan et al. Science paper, it would be appropriate to bring the rebuttal of their findings in Öktem et al. (*JAS* 2023). Just to show that the science is not as obvious as Fan et al. imply.

4. L. 74: "Numerous other modeling studies…". L. 92: "…despite numerous studies supporting the idea…". Such statements make me believe that the authors do believe in the invigoration, and do not seriously consider those who base their convictions on physical processes and object simple explanations of the "observed" impact of pollution on convection. What about studies that provide picture consistent with theoretical considerations as in Grabowski and Morrison papers or apply more careful analysis of observations (e.g., Varble *JAS* 2018, Öktem et al. *JAS* 2023)?

5. K-index, LR, and many other acronyms. All those should be defined once first used in the text. The reference to Table 1 that explains those acronyms in more detail should be included when these are introduced. I have to admit that, for someone not familiar with airborne instrumentation, the number of acronyms used in the paper is frustrating. As an example, QC for "quality control" is used just 4 times, and I do not think the acronym is needed.

6. L. 136: The reference to Kretschmer et al. is not needed. That paper is from a different field and not really explaining the difference between correlation and causality. The fact that correlation does not imply causality should be obvious in its own right, and it is recognized by some papers the authors cite (e.g., see the abstract in Lin et al. *JGR* 2006 already cited in the manuscript).

7. Text references to entries in the Table 1 can be improved. For instance, L.288: I suggest "…AVAPS parameters marked as "environmental" in Table 1 were employed…". L. 297: "…marked as "aerosol" in Table 1".

8. L. 329: "…result was unexpected…". Why? See my comments above.

References not cited in the manuscript:

Andreae, M. O., Rosenfeld, D., Artaxo, P., Costa, A. A., Frank, G. P., Longo, K. M., and Silva-Dias, M. A.: Smoking rain clouds over the Amazon, Science, 303, 1337–1342, https://doi.org/10.1126/science.1092779, 2004.

Öktem, R., D. M. Romps, and A. C. Varble, 2023: No Warm-Phase Invigoration of Convection Detected during GoAmazon. J. Atmos. Sci., 80, 2345–2364, https://doi.org/10.1175/JAS-D-22-0241.1.

Varble, A., 2018: Erroneous Attribution of Deep Convective Invigoration to Aerosol Concentration. J. Atmos. Sci., 75, 1351–1368, https://doi.org/10.1175/JAS-D-17-0217.1.

Signed: W. Grabowski.

---

## Referee Report (RR2)

2nd Review of *Observed impacts of aerosol concentration on maritime tropical convection within constrained environments using airborne radiometer, radar, lidar, and dropsondes* by Amiot et al. (2024).

I commend the authors for making significant methodological improvements in their manuscript. Indeed, it appears that these changes had a substantial impact on some of the correlations and I believe that the results should be more robust now. Nevertheless, I still find some major problems with the manuscript and I suggest to return the manuscript to the authors for ***major revisions***.

Please note that line numbers are based on the manuscript without tracked changes.

**General comments**

1. My most major concern regards the overall interpretation of the results. Despite the more robust methodological approach, there are still many unexpected results. Furthermore, if one investigates correlations for such a large number of variables one is bound to find a correlation somewhere. What about all the non-correlated examples? There is little to no discussion about them. Yes, there are some potentially interesting correlations that the authors can report, but I still feel that these correlations are strongly influenced by one or two outliers. Take these points away and the correlation disappears. Visually it seems that correlations could be entirely different in most cases if just a few points are changed. Are the outliers really related to aerosol-cloud interactions or is there something entirely different going on? To me the authors have not sufficiently demonstrated that different processes can be ruled out (see for example comment 4).

   In fact, in Figures 5 and 7 there appears to be a general trend that for larger sample sizes correlations are smaller. Below I also show an example from Figure 7. For 850-500 LR there is a strong correlation in the medium bin, but for the 700-500 LR there is strong correlation in the high and low bins. It seems likely that this happens because of just a few points shifting between bins.

   | 700-500 LR | | | | | | |
   |---|---|---|---|---|---|---|
   | 0.78 (9) | 0.74 (9) | 0.82 (9) | 0.83 (9) | 0.79 (9) | 0.69 (9) | H |
   | 0.09 (10) | 0.04 (10) | 0.16 (10) | 0.17 (10) | 0.12 (10) | 0.16 (10) | M |
   | 0.90 (8) | 0.80 (8) | 0.96 (7) | 0.96 (8) | 0.82 (9) | 0.92 (9) | L |
   | 0.28 (9) | 0.27 (9) | 0.41 (9) | 0.43 (9) | 0.45 (9) | 0.53 (9) | H |
   | 0.83 (9) | 0.81 (9) | 0.84 (9) | 0.84 (9) | 0.79 (9) | 0.72 (9) | M |
   | 0.07 (9) | -0.04 (9) | 0.07 (8) | 0.01 (9) | 0.06 (10) | 0.07 (10) | L |

   850-500 LR

   To me personally the results remain rather inconclusive. Maybe the authors can rethink their interpretation of the data by emphasizing inconclusiveness due to

limited samples and many unexpected trends. I do believe that such results should be reported as well. The way the results are currently reported I find it hard to justify publication in ACP. We have seen in previous literature that sometimes there are correlations between convective parameters and aerosol measurements. Is it really surprising to find this in a new dataset when looking at 100s of potential correlations?

2. Maybe a bootstrapping approach could help investigate the issue of outliers. Randomly remove 10 – 20% of the data points many times and recalculate correlations to achieve a mean correlation. For the small sample sizes this might need to be limited to unique realizations of the dataset. However, I fear that overall correlations would become much weaker.

3. The introduction has improved significantly in terms of the covered literature. However, I still feel it is not very engaging, mostly it is just listing previous research without telling a story that motivates the research. Still the paragraphs starting in lines 42 and 103 seem out of place to me. One part that is missing is why we care about aerosol impacts on convection. It is interesting that there are interactions but what are the potential consequences (radiation, precipitation, etc.)? I think that is what the introduction should start with.

4. I am unsatisfied with the answer to my previous comment 4 about different convective regimes. I do believe that these can significantly impact results since development mechanisms will differ between types of convection. The authors mentioned a squall line. More organized convection might have developed 100s of km away in a different aerosol and thermodynamic environment. Maybe some shallow cumulus clouds could be invigorated compared to other shallow cumulus clouds, but we cannot really see this because all shallow cumulus will appear as weak convection compared to deeper cumulus. Just to name a couple of problems. In essence, this introduces a lot of noise to the results which might be causing some of the outliers that appear to be causing many of the trends.

5. I am also still unsatisfied with the description of the CAMP2Ex campaign. Can the authors please make a dedicated section about the campaign in the methods section? In general, the methods section would benefit from some dedicated subsections so it is easier to find specific details about the methodology.

**Specific comments**

6. 57: Probably better to use 'clouds' here instead of storms.
7. 59: 'describes' instead of 'favors'.
8. Figure 1: Excuse me if I do not fully understand the measurements, but why is there significant CLW where the radar does not observe any cloud?
9. 127: These variables have not been defined.
10. 157: 'absolute deviation', does this refer to CLW?
11. 262: I don't quite understand why scenes longer than 11 minutes had to be masked. Couldn't you just use the 10 minutes around the dropsondes?
12. 347-350: Can points with significant masking be indicated somehow? Maybe use the same symbol but only show the outline for scenes where masking exceeded a certain threshold.
13. 549-551: I think I understand what the authors are trying to say, but please consider rephrasing.
14. 585: The following paragraph is missing some kind of discussion about why at least some of the things mentioned here were not done in this study.

---

## Referee Report (RR3)

**Review of *Observed impacts of aerosol concentration on maritime tropical convection within constrained environments using airborne radiometer, radar, lidar, and dropsondes* by Amiot et al.**

**Recommendation: *Publish subject to minor revisions***

The authors have provided a much-improved manuscript based on the reviewers' comments. Especially, the introduction and methodology section are now much more suitable for publication. Overall, I am satisfied with the responses. I think the writing could be tightened up in certain parts. For example, there is a quite extensive description of results in the supplementary material (466-483). This part could be shortened quite a bit. In general, I suggest carefully reading the manuscript and editing it to shorten the text.

Below, I list a few more comments that the authors may want to consider before publication. These are mostly suggestions and are up to the authors if they want to include them.

**Comments**

1. 129: 'radar-based proxy'
2. 416: 'is capturing'
3. Can the corresponding parts of Figure 2 shown in Figure 3 be indicated somehow? For example, by putting a thicker box in Figure 2 around the three values for Mean 355-nm Bsc binned by LCL shown in Figure 3a. The same can be done for Figures 4,6, and 7. That should make it easier to find the parts of the figures that correspond to each other.
4. In general, I would suggest flipping x-axis and y-axis in Figures 3,5, and 8 since technically the authors are investigating the convective variables as a function of the aerosol variables.
5. 525: K-index looks like it has the lowest correlations in Fig. 6.

---

## Author Response (AR2)

**Responses to Reviewers**

We would like to thank the Reviewers for their time in reviewing the revised version of our manuscript and for the thoughtful feedback they have provided. As discussed in greater detail in our point-by-point responses below, we have incorporated your suggestions in the newly updated version of the manuscript. Compared to the previous version, the most substantial changes include:

- An overhaul of the Introduction section to convey the background and motivation for our study with a clearer story and better representation of the conclusions in prior literature.
- Breaking section 2 apart into five subsections, including a more comprehensive overview of CAMP$^2$Ex in section 2.1.
- Moving the AMPR cloud liquid water (CLW) analysis from section 3 to supplemental material. We have decided that AMPR's polarization-corrected temperatures (PCTs) should be the primary AMPR results as they are available in regions of strongest precipitation sampled, which is a key indirect indicator of convective intensity in this study. In contrast, as was previously discussed in the manuscript, AMPR's CLW retrievals fail in regions of moderate-to-high precipitation. While we had hypothesized that key trends would be observable in these results, we feel that the CLW results distract from the main point of focusing on regions of strongest sampled convection, especially since the results sections opened with the CLW analysis in the previous manuscript revision. Therefore, while we want to present the CLW results, we feel they are better suited for supplemental material with some key references made to them in the main text.
- Removing 700-hPa vertical velocity ($w_{700}$) as an environmental parameter. This was based on questions regarding the true magnitude of the $w_{700}$ values observed compared to the uncertainty in this derived product from the dropsondes. Ultimately, while interesting trends may be discernable from this product, we are less confident in it compared to the other environmental parameters and feel it would be best to exclude it from the manuscript.
- Including a bootstrapping analysis, with all results in sections 3 and 4 now reflecting mean values calculated across the 1000 runs used in this analysis.
- Giving greater attention to some of the more-inconclusive results from our study, including a general shift in the conversation throughout sections 3–5 to better convey some of these inconclusive results rather than focusing on the strongest correlations (though the latter discussions are still present in sections 3–5).
- The identification and correction of two primary coding errors made by the lead author. These errors involved cases where some of the APR-3 files were mismatched with the AMPR and HSRL2 data in a minority of the scenes examined in our study. Ultimately, while some of the values reported in the correlation analyses changed because of these corrections, the science was not greatly impacted and the overall message in our study did not change due to these corrections. This is especially true since we have now given greater focus on the general inconclusiveness indicated by many of our analyses.

Our responses to specific Reviewer comments can be found below, wherein Reviewer comments are presented in italicized font and our response immediately follows in standard font.

**Reviewer #1:**

*Review of revised manuscript "Observed impacts of aerosol concentration on maritime tropical convection within constrained environments using airborne radiometer, radar, lidar, and dropsondes" by Amiot et al.*

*Recommendation: accept after revisions*

*Overall evaluation: I think the paper has improved. Some of my comments on the observational uncertainty and interpretation of the observations (e.g., correlation versus causation) are now addressed (at least as far as I can tell). However, I claim that the introduction and motivation for the study presents an unclear (or even misleading) picture of the invigoration conundrum. In my opinion, the introduction is written from the perspective of a person who believes that the convective invigoration in polluted environments (all other factor being equal) is a proven effect. On physical grounds, the pollution-induced invigoration has little merit, especially the so-called cold invigoration that strongly depends on details of the freezing and condensate off-loading aloft (see section 2 in Grabowski and Morrison JAS 2020 and discussions in Igel and van den Heever GRL 2021 and Varble et al. ACP 2023). Below I provide several specific comments that the authors should address before the manuscript is accepted.*

We thank you very much for your review of our revised manuscript, and we have included your recommendations in the latest version as discussed in our responses below.

*Specific comments on the introduction.*

*The introduction of the pollution-induced invigoration in the third paragraph of the opening section should start with two seminal papers that initiated the discussion: Andreae et al. (Science 2004, not listed in the manuscript) and Rosenfeld et al. (Science 2008). The brief review of studies trying to prove and disprove the impact in observations and modeling should follow. However, the physical basis of the invigoration should be also brought into the picture following the discussion in the papers listed above. It would be appropriate to point out that the warm-phase invigoration depends on the finite supersaturation within cloud updrafts (because reducing supersaturation in polluted clouds increases buoyancy), with the condensation rate depending only on the updraft velocity as long as the supersaturation is equal to the quasi-equilibrium supersaturation (see section 2b in Grabowski and Morrison JAS 2020). The cold-phase invigoration critically depends on the details of the frozen precipitation off-loading aloft because the latent heating due to freezing approximately balances the weight of the liquid water carried across the melting level (see section 2a in Grabowski and Morrison as well as Igel and van den Heever and Varble et al. papers). I feel a correct motivation for the study under review is important for providing a proper context. It also reflects in my view rather mixed results discussed in the paper, perhaps because details of the physical mechanisms involved (finite supersaturations, freezing and off-loading cloud condensate aloft) are practically impossible to document in observations. However, I feel the study is an important contribution to the problem, but a proper perspective of the past research is important.*

We have revised the content and layout of the introduction (section 1) based on your suggestions and those of the other Reviewer. The updated introduction and motivation follow your suggested format/flow, including the references you've listed here and at the end of your review.

The general layout of section 1 is now:

1. Describe the purpose and importance of our study
2. Summarize the secondary indirect effect of aerosols
3. Introduce the physics behind warm- and cold-phase invigoration
4. List some example studies whose results support these invigoration mechanisms
5. List some example studies whose results counter these invigoration mechanisms
6. Use the mixed results from these studies to springboard our study and hypotheses, including more in-depth discussions about the relationships between convective intensity and the microwave remote sensing signatures we've examined

*Specific detailed comments.*

*1.      L. 44: What do you mean by "environmental contexts". Please explain.*

This phrase was meant to indicate that we would consider the environmental conditions around observed aerosol and convective metrics when discussing the implications of possible aerosol-cloud interactions within a given scene. We have modified this wording on line 44 to be "… while considering adjacent environmental conditions."

*2.      L. 65. The warm-phase invigoration is not defined. L. 91: the same for cold-phase invigoration.*

These concepts are now defined in the introduction on lines 61–65.

*3.      When you bring Fan et al. Science paper, it would be appropriate to bring the rebuttal of their findings in Öktem et al. (JAS 2023). Just to show that the science is not as obvious as Fan et al. imply.*

We have added the study by Öktem et al., (2023), including its contradiction to the results of Fan et al., (2018) on lines 101–102.

*4.      L. 74: "Numerous other modeling studies…". L. 92: "…despite numerous studies supporting the idea…". Such statements make me believe that the authors do believe in the invigoration, and do not seriously consider those who base their convictions on physical processes and object simple explanations of the "observed" impact of pollution on convection. What about studies that provide picture consistent with theoretical considerations as in Grabowski and Morrison papers or apply more careful analysis of observations (e.g., Varble JAS 2018, Öktem et al. JAS 2023)?*

We appreciate the Reviewer pointing this out, as our wording was meant to indicate the prevalence of literature that support the aerosol invigoration of convection.  We have modified this wording to present the differing conclusions of prior studies more clearly.  Specifically, we now discuss (lines 75–92) how some studies have proposed and/or supported the idea of aerosol invigoration of convection while balancing this with a similar-length discussion (lines 93–114) of studies whose conclusions generally don't support the aerosol invigoration of convection, including the manuscripts you've suggested.

*5.      K-index, LR, and many other acronyms. All those should be defined once first used in the text. The reference to Table 1 that explains those acronyms in more detail should be included when these are introduced. I have to admit that, for someone not familiar with airborne instrumentation, the number of acronyms used in the paper is frustrating. As an example, QC for "quality control" is used just 4 times, and I do not think the acronym is needed.*

We have now used more-general descriptions and terminology when introducing our hypotheses on lines 115–156, and we return to these hypotheses at the end of section 2 (i.e., lines 388–392) to list specific expectations for each convective and environmental parameter after defining their acronyms throughout section 2 and describing them in greater detail therein.  We have also removed several acronyms from the updated manuscript, now spelling out their full names during their relatively infrequent usage, including: cloud condensation nuclei (CCN), hydrometeor diameter (D), equilibrium level (EL), level of free convection (LFC), noise-equivalent differential temperature (NEDT), quality control (QC), relative humidity (RH), and science flight (SF).

*6.      L. 136: The reference to Kretschmer et al. is not needed. That paper is from a different field and not really explaining the difference between correlation and causality. The fact that correlation does not imply causality should be obvious in its own right, and it is recognized by some papers the authors cite (e.g., see the abstract in Lin et al. JGR 2006 already cited in the manuscript).*

We have removed the reference to Kretschmer et al. (2017) from the manuscript as you've suggested, and we've referenced Lin et al. (2006) on line 148 when noting differences between correlation and causality.

*7.	Text references to entries in the Table 1 can be improved. For instance, L.288: I suggest "...AVAPS parameters marked as "environmental" in Table 1 were employed…". L. 297: "...marked as "aerosol" in Table 1".*

We have provided additional details when referencing Table 1 in the manuscript text, especially around lines 352–356.

*8.	L. 329: "...result was unexpected…". Why? See my comments above.*

We have removed this statement from the manuscript to coincide with the modifications made to our introduction section.

*References not cited in the manuscript:*

*Andreae, M. O., Rosenfeld, D., Artaxo, P., Costa, A. A., Frank, G. P., Longo, K. M., and Silva-Dias, M. A.: Smoking rain clouds over the Amazon, Science, 303, 1337–1342, https://doi.org/10.1126/science.1092779, 2004.*

*Öktem, R., D. M. Romps, and A. C. Varble, 2023: No Warm-Phase Invigoration of Convection Detected during GoAmazon. J. Atmos. Sci., 80, 2345–2364, https://doi.org/10.1175/JAS-D-22-0241.1.*

*Varble, A., 2018: Erroneous Attribution of Deep Convective Invigoration to Aerosol Concentration. J. Atmos. Sci., 75, 1351–1368, https://doi.org/10.1175/JAS-D-17-0217.1.*

*Signed: W. Grabowski.*

**Reviewer #2:**

*2nd Review of Observed impacts of aerosol concentration on maritime tropical convection within constrained environments using airborne radiometer, radar, lidar, and dropsondes by Amiot et al. (2024).*

*I commend the authors for making significant methodological improvements in their manuscript. Indeed, it appears that these changes had a substantial impact on some of the correlations and I believe that the results should be more robust now. Nevertheless, I still find some major problems with the manuscript and I suggest to return the manuscript to the authors for **major revisions**.*

*Please note that line numbers are based on the manuscript without tracked changes.*

We thank you very much for your review of our revised manuscript, and we have incorporated your suggestions into the updated version as noted in our responses below.

***General comments***

*1.      My most major concern regards the overall interpretation of the results. Despite the more robust methodological approach, there are still many unexpected results. Furthermore, if one investigates correlations for such a large number of variables one is bound to find a correlation somewhere. What about all the non-correlated examples? There is little to no discussion about them. Yes, there are some potentially interesting correlations that the authors can report, but I still feel that these correlations are strongly influenced by one or two outliers. Take these points away and the correlation disappears. Visually it seems that correlations could be entirely different in most cases if just a few points are changed. Are the outliers really related to aerosol-cloud interactions or is there something entirely different going on? To me the authors have not sufficiently demonstrated that different processes can be ruled out (see for example comment 4).*

*In fact, in Figures 5 and 7 there appears to be a general trend that for larger sample sizes correlations are smaller. Below I also show an example from Figure 7. For 850-500 LR there is a strong correlation in the medium bin, but for the 700-500 LR there is strong correlation in the high and low bins. It seems likely that this happens because of just a few points shifting between bins.*

| | | | | | | |
|---|---|---|---|---|---|---|
| 0.78 (9) | 0.74 (9) | 0.82 (9) | 0.83 (9) | 0.79 (9) | 0.69 (9) | H |
| 0.09 (10) | 0.04 (10) | 0.16 (10) | 0.17 (10) | 0.12 (10) | 0.16 (10) | M |
| 0.90 (8) | 0.80 (8) | 0.96 (7) | 0.96 (8) | 0.82 (9) | 0.92 (9) | L |
| 0.28 (9) | 0.27 (9) | 0.41 (9) | 0.43 (9) | 0.45 (9) | 0.53 (9) | H |
| 0.83 (9) | 0.81 (9) | 0.84 (9) | 0.84 (9) | 0.79 (9) | 0.72 (9) | M |
| 0.07 (9) | -0.04 (9) | 0.07 (8) | 0.01 (9) | 0.06 (10) | 0.07 (10) | L |

*(left axis labels: 700-500 LR for top three rows, 850-500 LR for bottom three rows)*

*To me personally the results remain rather inconclusive. Maybe the authors can rethink their interpretation of the data by emphasizing inconclusiveness due to limited samples and many unexpected trends. I do believe that such results should be reported as well. The way the results are currently reported I find it hard to justify publication in ACP. We have seen in previous literature that sometimes there are correlations between convective parameters and aerosol measurements. Is it really surprising to find this in a new dataset when looking at 100s of potential correlations?*

We have significantly changed sections 3–5 in the updated manuscript, including a complete overhaul of sections 3 and 4, in response to the concerns you have raised and to reflect the updated results after correcting the coding errors listed on page 1.  Given the changes to the correlation tables, in addition to the points you've raised about the results in general, we now provide much greater focus on the general inconclusive nature of the results in our discussions throughout sections 3–5 and in the abstract.  In addition to noting and discussing the presence of numerous weak and/or statistically insignificant correlations resulting from our analyses, we specifically note and highlight some of the more-unexpected trends in our discussions of the scatterplots.  Further, Fig. 8a is devoted to providing an in-depth examination of an aerosol-convective comparison wherein the correlations were weak and statistically insignificant for all three environmental bins, whereas our previous manuscript versions had solely focused on producing scatterplots to include some of the strongest positive correlations with high statistical significance.  We do still highlight some of the strongest and/or most statistically significant correlations in the scatterplots, but we take greater care to present these as potentially interesting trends that may be worthy of future analysis while acknowledging that they appear among a greater abundance of weaker and less statistically significant correlations in our results.

Regarding the figure segment you've included in your review, we wanted to mention that the effect of "a few points shifting between bins" cannot be ascertained by comparing the 850–500-hPa LR and 700–500-hPa LR values in (the former) Fig. 7 since these represent two different, albeit similar, environmental conditions.  To examine the effects of data points shifting between the low-medium-high bins, the sensitivity tests in (the former) Fig. S10 must be examined, wherein most of the correlation trends did not change significantly.  However, in agreement with your point, there is indeed sensitivity amongst the correlation values depending on how many data points fall into each bin and what the low-medium and medium-high thresholds are.  There are also cases where the correlations do, ultimately, change considerably among the sensitivity tests as you've noted, especially for very small samples (e.g., the high category of 700–500-hPa LR, which always contained fewer than 10 data points in the sensitivity tests shown in the previous manuscript version).

*2.       Maybe a bootstrapping approach could help investigate the issue of outliers. Randomly remove 10 – 20% of the data points many times and recalculate correlations to achieve a mean correlation. For the small sample sizes this might need to be limited to unique realizations of the dataset. However, I fear that overall correlations would become much weaker.*

This is an interesting idea.  In the updated manuscript, we have utilized a bootstrapping approach for all correlations and p-values investigated in our study, wherein 10% of the paired convective-aerosol data array elements (rounded up to the nearest whole integer) were withheld.  A new correlation coefficient and p-value were calculated using these "reduced" arrays, and this process was repeated 1000 times for each paired convective-aerosol data array before the resulting 1000 correlation coefficients and p-values were each averaged.  This procedure has been added to the manuscript on lines 362–372.  In addition, as noted on lines 367–368, the values in Figs. 2, 4, 6, and 7 now reflect these mean correlation coefficient values, the associated mean p-values, and the mean number of data points considered during this bootstrapping approach for each comparison.

Most of these newly calculated mean correlation coefficients and p-values differed very little compared to the previous non-bootstrapped analysis.  An example of these differences for the AMPR CLW and Pixels$_{Ku}$ analyses is provided in Fig. R1 on the next page.  Most correlations were within 0.01 of their previous values in each figure.  There were some slight (i.e., ~0.01–0.02) increases in the associated p-values that resulted from averaging across the 1000 comparisons, which appear a bit more striking in the figures given the color/shading gradient in the selected colorbar.  However, the "most-significant" correlations with a p-value < 0.01 typically retained a fairly low p-value near 0.01–0.03.  As you noted, the bootstrapping approach posed some challenges for small array sizes, namely that any comparisons with a sample size < 10 were left unchanged since removing 10% of the dataset and rounding up to the nearest whole integer yielded the same sample size.  The largest changes occurred for sample sizes of 10–15, where removing a single data point had a relatively strong impact on the calculated correlations, which matches expectations.  We have added descriptions of the bootstrapping results throughout the discussions in sections 3 and 4 based on the new values in Figs. 2, 4, 6, and 7.

**Figure: AMPR CLW — prior to applying the bootstrapping approach (top-left)**

[revised manuscript text omitted]

**Figure: Pixels$_{Ku}$ — prior to applying the bootstrapping approach (bottom-left)**

| | 355-nm AOT | 532-nm AOT | 355-nm Ext | 532-nm Ext | 355-nm Bsc | 532-nm Bsc | |
|---|---|---|---|---|---|---|---|
| $T_d$ 1 km | 0.44 (22) | 0.43 (23) | 0.43 (21) | 0.41 (23) | 0.36 (24) | 0.42 (25) | H |
| | 0.06 (26) | 0.06 (27) | 0.11 (26) | 0.15 (27) | 0.21 (28) | 0.07 (29) | M |
| | 0.17 (12) | 0.22 (13) | 0.19 (12) | 0.25 (13) | 0.25 (12) | 0.34 (13) | L |
| $T_d$ 925 hPa | 0.08 (21) | 0.08 (23) | 0.10 (20) | 0.09 (23) | 0.09 (23) | 0.15 (25) | H |
| | 0.12 (27) | 0.13 (27) | 0.16 (27) | 0.19 (27) | 0.22 (28) | 0.15 (28) | M |
| | 0.41 (12) | 0.44 (13) | 0.45 (12) | 0.49 (13) | 0.51 (13) | 0.59 (14) | L |
| 700-500 LR | 0.78 (9) | 0.74 (9) | 0.82 (9) | 0.83 (9) | 0.79 (9) | 0.69 (9) | H |
| | 0.09 (10) | 0.04 (10) | 0.16 (10) | 0.17 (10) | 0.12 (10) | 0.16 (10) | M |
| | 0.90 (8) | 0.80 (8) | 0.96 (7) | 0.96 (8) | 0.82 (9) | 0.92 (9) | L |
| 850-500 LR | 0.28 (9) | 0.27 (9) | 0.41 (9) | 0.43 (9) | 0.45 (9) | 0.53 (9) | H |
| | 0.83 (9) | 0.81 (9) | 0.84 (9) | 0.84 (9) | 0.79 (9) | 0.72 (9) | M |
| | 0.07 (9) | -0.04 (9) | 0.07 (8) | 0.01 (9) | 0.06 (10) | 0.07 (10) | L |
| 850-700 LR | 0.09 (20) | 0.13 (23) | 0.11 (20) | 0.15 (23) | 0.16 (20) | 0.19 (23) | H |
| | 0.33 (16) | 0.31 (16) | 0.42 (16) | 0.42 (16) | 0.42 (16) | 0.46 (16) | M |
| | -0.07 (24) | -0.07 (24) | -0.10 (24) | -0.12 (24) | -0.26 (26) | -0.17 (26) | L |
| K-Index | 0.22 (10) | 0.18 (10) | 0.38 (9) | 0.40 (10) | 0.48 (10) | 0.47 (10) | H |
| | 0.96 (9) | 0.94 (9) | 0.94 (9) | 0.93 (9) | 0.91 (10) | 0.89 (10) | M |
| | 0.31 (8) | 0.22 (8) | 0.38 (8) | 0.39 (8) | 0.41 (8) | 0.15 (8) | L |
| LCL Alt. | 0.17 (22) | 0.17 (22) | 0.17 (22) | 0.18 (22) | 0.29 (24) | 0.24 (24) | H |
| | 0.20 (20) | 0.23 (21) | 0.24 (20) | 0.28 (21) | 0.25 (21) | 0.24 (22) | M |
| | 0.53 (18) | 0.45 (20) | 0.57 (17) | 0.57 (20) | 0.45 (19) | 0.46 (21) | L |
| NCAPE | 0.14 (27) | 0.17 (29) | 0.17 (26) | 0.20 (29) | 0.19 (27) | 0.27 (29) | H |
| | -0.17 (13) | -0.32 (13) | -0.09 (13) | -0.01 (13) | 0.49 (14) | -0.05 (14) | M |
| | 0.12 (20) | 0.15 (21) | 0.19 (20) | 0.25 (21) | 0.29 (23) | 0.12 (24) | L |
| 700 hPa w | 0.18 (23) | 0.21 (24) | 0.20 (23) | 0.23 (24) | 0.12 (23) | 0.19 (24) | H |
| | 0.19 (16) | 0.20 (17) | 0.23 (16) | 0.26 (17) | 0.29 (18) | 0.28 (19) | M |
| | 0.24 (21) | 0.24 (22) | 0.24 (20) | 0.24 (22) | 0.26 (21) | 0.30 (22) | L |
| | 355-nm AOT | 532-nm AOT | 355-nm Ext | 532-nm Ext | 355-nm Bsc | 532-nm Bsc | |

**Figure: Pixels$_{Ku}$ — after applying the bootstrapping approach (bottom-right)**

| | 355-nm AOT | 532-nm AOT | 355-nm Ext | 532-nm Ext | 355-nm Bsc | 532-nm Bsc | |
|---|---|---|---|---|---|---|---|
| $T_d$ 1 km | 0.42 (20) | 0.42 (21) | 0.43 (19) | 0.40 (21) | 0.36 (22) | 0.41 (23) | H |
| | 0.07 (24) | 0.06 (25) | 0.11 (24) | 0.15 (25) | 0.22 (26) | 0.07 (27) | M |
| | 0.17 (11) | 0.22 (12) | 0.19 (11) | 0.25 (12) | 0.24 (11) | 0.34 (12) | L |
| $T_d$ 925 hPa | 0.08 (19) | 0.08 (21) | 0.10 (18) | 0.08 (21) | 0.08 (21) | 0.14 (23) | H |
| | 0.12 (25) | 0.13 (25) | 0.17 (25) | 0.18 (25) | 0.21 (26) | 0.15 (26) | M |
| | 0.44 (11) | 0.46 (12) | 0.47 (11) | 0.50 (12) | 0.52 (12) | 0.60 (13) | L |
| 700-500 LR | 0.78 (9) | 0.74 (9) | 0.82 (9) | 0.83 (9) | 0.79 (9) | 0.69 (9) | H |
| | 0.12 (9) | 0.07 (9) | 0.15 (9) | 0.18 (9) | 0.13 (9) | 0.15 (9) | M |
| | 0.90 (8) | 0.80 (8) | 0.96 (7) | 0.96 (8) | 0.82 (9) | 0.92 (9) | L |
| 850-500 LR | 0.28 (9) | 0.27 (9) | 0.41 (9) | 0.43 (9) | 0.45 (9) | 0.53 (9) | H |
| | 0.83 (9) | 0.81 (9) | 0.84 (9) | 0.84 (9) | 0.79 (9) | 0.72 (9) | M |
| | 0.07 (9) | -0.04 (9) | 0.07 (8) | 0.01 (9) | 0.08 (9) | 0.09 (9) | L |
| 850-700 LR | 0.07 (18) | 0.10 (21) | 0.08 (18) | 0.13 (21) | 0.15 (18) | 0.16 (21) | H |
| | 0.34 (15) | 0.32 (15) | 0.43 (14) | 0.43 (15) | 0.43 (15) | 0.46 (15) | M |
| | -0.08 (22) | -0.07 (22) | -0.10 (22) | -0.12 (22) | -0.26 (24) | -0.17 (24) | L |
| K-Index | 0.22 (9) | 0.18 (9) | 0.38 (9) | 0.39 (9) | 0.48 (9) | 0.43 (9) | H |
| | 0.96 (9) | 0.94 (9) | 0.94 (9) | 0.93 (9) | 0.86 (9) | 0.86 (9) | M |
| | 0.31 (8) | 0.22 (8) | 0.38 (8) | 0.39 (8) | 0.41 (8) | 0.15 (8) | L |
| LCL Alt. | 0.16 (20) | 0.17 (20) | 0.17 (20) | 0.18 (20) | 0.29 (22) | 0.23 (22) | H |
| | 0.20 (18) | 0.24 (19) | 0.25 (18) | 0.28 (19) | 0.25 (19) | 0.24 (20) | M |
| | 0.52 (17) | 0.41 (18) | 0.55 (16) | 0.54 (18) | 0.44 (18) | 0.43 (19) | L |
| NCAPE | 0.14 (25) | 0.17 (27) | 0.17 (24) | 0.20 (27) | 0.19 (25) | 0.27 (27) | H |
| | -0.17 (12) | -0.31 (12) | -0.09 (12) | -0.01 (12) | 0.49 (13) | -0.04 (13) | M |
| | 0.11 (18) | 0.15 (19) | 0.19 (18) | 0.24 (19) | 0.30 (21) | 0.12 (22) | L |
| 700 hPa w | 0.18 (21) | 0.21 (22) | 0.20 (21) | 0.23 (22) | 0.12 (21) | 0.19 (22) | H |
| | 0.19 (15) | 0.20 (16) | 0.24 (15) | 0.26 (16) | 0.28 (17) | 0.28 (18) | M |
| | 0.23 (19) | 0.24 (20) | 0.23 (20) | 0.25 (19) | 0.29 (20) | 0.29 (20) | L |
| | 355-nm AOT | 532-nm AOT | 355-nm Ext | 532-nm Ext | 355-nm Bsc | 532-nm Bsc | |

**Figure R1: Tables of Pearson correlation coefficients and their statistical significances, as in Fig. S4 of the supplemental material. Herein, the top row presents AMPR CLW and the bottom row presents Pixels$_{Ku}$ prior to (left) and after (right) applying the bootstrapping approach. These figures are based on the previous version of the manuscript and before any adjustments were made to the Python codes as outlined on page 1 of this document.**

*3.        The introduction has improved significantly in terms of the covered literature. However, I still feel it is not very engaging, mostly it is just listing previous research without telling a story that motivates the research. Still the paragraphs starting in lines 42 and 103 seem out of place to me. One part that is missing is why we care about aerosol impacts on convection. It is interesting that there are interactions but what are the potential consequences (radiation, precipitation, etc.)? I think that is what the introduction should start with.*

We have made additional changes to the introduction (section 1) in the updated manuscript based on your suggestions.  The first paragraph of the introduction now opens with the purpose and a note about using CAMP$^2$Ex data, pointing the reader to section 2 for a more-thorough overview of the campaign, before moving into a description of some consequences of the aerosol impacts on convection from a meteorological perspective and the broader societal impacts.  The paragraphs that began on lines 42 and 103 in the previous manuscript version have had much of their content moved to section 2.  In addition, following the suggestions of the other Reviewer, we have modified the introduction to provide more of a comprehensive overview of different results from prior literature and have reworked several paragraphs to create a more cohesive narrative as motivation for our study.

The general layout of section 1 is now:

1.  Describe the purpose and importance of our study
2.  Summarize the secondary indirect effect of aerosols
3.  Introduce the physics behind warm- and cold-phase invigoration
4.  List some example studies whose results support these invigoration mechanisms
5.  List some example studies whose results counter these invigoration mechanisms
6.  Use the mixed results from these studies to springboard our study and hypotheses, including more in-depth discussions about the relationships between convective intensity and the microwave remote sensing signatures we've examined

*4.        I am unsatisfied with the answer to my previous comment 4 about different convective regimes. I do believe that these can significantly impact results since development mechanisms will differ between types of convection. The authors mentioned a squall line. More organized convection might have developed 100s of km away in a different aerosol and thermodynamic environment. Maybe some shallow cumulus clouds could be invigorated compared to other shallow cumulus clouds, but we cannot really see this because all shallow cumulus will appear as weak convection compared to deeper cumulus. Just to name a couple of problems. In essence, this introduces a lot of noise to the results which might be causing some of the outliers that appear to be causing many of the trends.*

We have performed an analysis where we have separated the observed clouds into two different classes, stratus and cumulus, throughout CAMP$^2$Ex and re-ran our analyses on these two groups of clouds separately.  When masking each column of APR-3 data according to these cloud classes, we used the following threshold values:

- Stratus: Ka-band 0-dBZ $Z_H$ contour extended vertically 1.98 km or less in the column
- Cumulus: Ka-band 0-dBZ $Z_H$ contour extended vertically 2.01 km or more in the column

These specific heights for the $Z_H$ contours were due to the 30-m gate spacing used in the APR-3 dataset. Likewise, we stratified the AMPR data according to:
- Stratus: CLW < 0.2 kg m$^{-2}$ in the column
- Cumulus: CLW ≥ 0.2 kg m$^{-2}$ in the column

Data columns wherein these conditions were not met were masked in their respective analysis. It should be noted that entire scenes were not necessarily masked, just the columns that were not associated with the cloud type of interest for the given analysis. We recognize that these values may not perfectly represent the clouds that would fall into each of these groups, but we found that they did a decent job of stratifying the datasets into two groups wherein a fairly significant number of correlations could be performed, especially since the bootstrapping methods noted in comment 2 above were employed in this analysis. We originally wanted to use more cloud classes (e.g., stratus, shallow cumulus, and deeper cumulus), but the sample proved too small to stratify into these three groups while maintaining a physical explanation behind each group (i.e., the original "shallow" cumulus cloud group required including clouds > 4.5 km tall based on the Ka-band 0-dBZ $Z_H$ contour to achieve a sample size large enough for most of the correlation analyses to be performed). The results of our analyses can be found in Figs. R2–R4 below.

In general, while the aerosol-convective correlations within each environmental group did change in response to isolating these two cloud classes, most did not change by a significant amount (i.e., cases where a correlation changed from moderately negative to moderately positive). Some comparisons did indeed see a considerable change in their correlation depending on which cloud class was examined, including cases where correlations were strong and statistically significant for one class but not the other. AMPR CLW was impacted most strongly, with widespread negative correlations for the stratus clouds but not cumulus clouds. This results from the clustering of data points around 0 kg m$^{-2}$ being included in the stratus-cloud analysis but not the cumulus-cloud analysis, similar to the statement made on line 476 in the manuscript.

We have decided not to include these results in the manuscript due to the severe limitations associated with this analysis, in addition to the limited sample size and limitations of the observational analysis discussed in the manuscript. In addition to the somewhat ad hoc thresholds used to separate the stratus and cumulus classes, this analysis cannot account for many other factors, such as whether the cumulus clouds developing, mature, or dissipating and, for squall lines, what the aerosol conditions were in the region wherein they initiated. We have also not considered the vertical distribution of aerosols in our study at all, which would significantly impact this cloud stratification (e.g., how the vertical aerosol distribution aligned with the position of a relatively thin stratus cloud). We wanted to include these results in our response to highlight them and demonstrate that we have considered them, but that we ultimately have chosen to maintain our grouping of all clouds together in the manuscript due to the limitations and caveats associated with this analysis. However, our CLW discussion in the manuscript now includes a discussion on lines 471–479 of the impacts very thin clouds with low CLW (e.g., < 1 g m$^{-2}$) have on the CLW-aerosol correlations, which are now presented in the supplemental material as mentioned on page 1.

[Figure]

**Figure R2: Tables of Pearson correlation coefficients and their statistical significances, as in Fig. S4 of the supplemental material. Herein, the top row presents AMPR CLW and the bottom row presents PCT$_{19}$ for the analysis of stratus clouds (left) and cumulus clouds (right), applying the cloud-type thresholds and bootstrapping approach discussed above. These figures are based on a semi-updated version of the manuscript after applying adjustments to the Python code outlined on page 1 of this document.**

[Figure]

**Figure R3: Tables of Pearson correlation coefficients and their statistical significances, as in Fig. S4 of the supplemental material. Herein, the top row presents $Z_{95,Ku}$ and the bottom row presents Pixels$_{Ku}$ for the analysis of stratus clouds (left) and cumulus clouds (right), applying the cloud-type thresholds and bootstrapping approach discussed above. These figures are based on a semi-updated version of the manuscript after applying adjustments to the Python code outlined on page 1 of this document.**

[Figure]

**Figure R4: Tables of Pearson correlation coefficients and their statistical significances, as in Fig. S4 of the supplemental material. Herein, DFR is presented for the analysis of stratus clouds (left) and cumulus clouds (right), applying the cloud-type thresholds and bootstrapping approach discussed above. These figures are based on a semi-updated version of the manuscript after applying adjustments to the Python code outlined on page 1 of this document.**

*5.    I am also still unsatisfied with the description of the CAMP2Ex campaign. Can the authors please make a dedicated section about the campaign in the methods section? In general, the methods section would benefit from some dedicated subsections so it is easier to find specific details about the methodology.*

We have broken the methods section into subsections 2.1–2.5.  Subsection 2.1 has been added to the manuscript as a subsection dedicated to describing the CAMP$^2$Ex field campaign in greater detail.

**Specific comments**

*6.    57: Probably better to use 'clouds' here instead of storms.*

We have simplified the wording "convective storms" to "convection" on line 44.

*7.    59: 'describes' instead of 'favors'.*

Modified as suggested.

*8.    Figure 1: Excuse me if I do not fully understand the measurements, but why is there significant CLW where the radar does not observe any cloud?*

Thank you for noting this.  We have identified a mismatch between the period covered by the APR-3 data in the scene figure compared to the period covered by the AMPR and HSRL2 data. Figure 1 has been updated to ensure all panels cover the same approximate 10-minute period. Due to the timing of the data reported by each instrument, including brief pauses (e.g., calibration scans), the exact times at each point along the x axis in Fig. 1 may not align completely across all panels; this can be seen, for example, by the x-axis offsets for the precipitating cloud observed around 0155 UTC.  However, the start and end times are approximately the same for all three instruments, and we have included time values along the x axes for the top three panels in Fig. 1.

*9.    127: These variables have not been defined.*

We have now used more-general descriptions and terminology when introducing our hypotheses on lines 115–156, and we return to these hypotheses at the end of section 2 (i.e., lines 388–392) to list specific expectations for each convective and environmental parameter after defining their acronyms throughout section 2 and describing them in greater detail therein.

*10.	157: 'absolute deviation', does this refer to CLW?*

Yes, it does; we have added this to line 198.

*11.	262: I don't quite understand why scenes longer than 11 minutes had to be masked. Couldn't you just use the 10 minutes around the dropsondes?*

This resulted from the manner in which the radar data were saved from CAMP$^2$Ex. For each science flight, the radar data were saved across several files with each file corresponding to a particular leg of the science flight. Because of this, there were cases where the 10-min scene around a given dropsonde covered more than one radar file / flight segment. The script used to match radar and dropsonde data was designed to identify the reported radar scan times nearest two times: (dropsonde launch time – 5 minutes) and (dropsonde launch time + 5 minutes). In some cases where the dropsonde was launched near the end of the time covered in a radar file, the next radar file did not start until several (i.e., > 5) minutes elapsed. The effect on our analysis was compounded by the masking we employed (e.g., aircraft maneuvers, excluding files wherein only W-band data were reported, etc.), even when stitching files together to create as seamless of a time series as possible. Because of this, the nearest radar scan to (dropsonde launch time + 5 minutes) may have actually been 10+ minutes later. Therefore, we mask scenes with times > 11 minutes as reported by this data-matching script. We similarly mask scenes where the time was < 9 minutes (e.g., a short < 10-minute flight segment between two aircraft maneuvers). We have added additional details about this data-masking method to the manuscript on lines 316–318.

*12.	347-350: Can points with significant masking be indicated somehow? Maybe use the same symbol but only show the outline for scenes where masking exceeded a certain threshold.*

We are concerned that the level of uncertainty in the masked data points is too large for their inclusion in the manuscript, even as outlined shaped within the (already fairly busy) figures. This is not solely due to native uncertainty within the retrievals, but the fact that their uncertainties become drastically higher within the data regions that were masked. In looking into your suggestion, we realized that additional details about some of the masks applied to the data in our study would be beneficial, which were outlined in Amiot, (2023) but not carried over to this manuscript. When looking at AMPR CLW in response to your question, the concern lies in the fact that a given AMPR data pixel was masked if one or more of the following were true: 1) the P-3 pitch and/or roll magnitude was ≥ 2°, since the retrievals are based on Earth incidence angle and assume level flight; 2) AMPR operated in a nadir-stare mode that was utilized for certain flight segments during CAMP$^2$Ex, for which the reliability of off-nadir pixels has not yet been evaluated; 3) the P-3 altitude was < 3 km AGL, as we noted issues with AMPR's data calibration due to insufficient cooling of its cold-load target at these lower altitudes during the

science flights; 4) the given AMPR scan included at least one pixel over land, as AMPR's signal is dominated by land emission over land rather than the geophysical parameters of interest to this study; and 5) precipitation was present within the pixel based on a $T_b$ thresholding method. We also mask the 10 pixels nearest the edges of AMPR's 50-pixel swath in each scan due to residual effects of a new radome that AMPR flew with during CAMP$^2$Ex. If a given pixel was flagged solely due to precipitation, which matches the discussion on lines 347–350 in the previous manuscript version, the uncertainty associated with the retrieved CLW therein would introduce additional noise to the scatterplots in this study. Because of this, we feel it is most important to exclude all of the masked data points from the scatterplots in this manuscript.

These flags are discussed in Lang et al., (2021) and Amiot, (2023), but we have added details about them to lines 187–190 for the reader's immediate reference.

*13.      549-551: I think I understand what the authors are trying to say, but please consider rephrasing.*

We have revised this sentence on lines 595–597.

*14.      585: The following paragraph is missing some kind of discussion about why at least some of the things mentioned here were not done in this study.*

We have added a brief discussion to many of the sentences in this paragraph (i.e., starting on line 632) to explain why these are suggested for future work rather than avenues we explored in this study (all of which were ultimately partially due to time constraints of the analysis presented in the manuscript). In particular, we note how we have looked at several convective, environmental, and aerosol metrics, and are suggesting others that could potentially be interesting to examine in a similar future study. We now note our emphasis on aerosol concentration and its relation to cloud particle size distribution, in addition to our decision to focus on P-3 instrumentation in this study. Lastly, we have removed the comment about examining scenes grouped into different types of convection, as we have now examined this as discussed in our reply to comment 4 above.

---

## Author Response (AR3)

**Responses to Reviewers**

We thank the two anonymous Reviewers for their suggested revisions to our manuscript. These recommendations have been incorporated into the updated manuscript as explained in our responses below. In these responses, the Reviewer comments are displayed in italicized font and our responses immediately follow in standard font.

**Reviewer #2**

*Review of Observed impacts of aerosol concentration on maritime tropical convection within constrained environments using airborne radiometer, radar, lidar, and dropsondes by Amiot et al.*

*Recommendation: Publish subject to minor revisions*

*The authors have provided a much-improved manuscript based on the reviewers' comments. Especially, the introduction and methodology section are now much more suitable for publication. Overall, I am satisfied with the responses. I think the writing could be tightened up in certain parts. For example, there is a quite extensive description of results in the supplementary material (466-483). This part could be shortened quite a bit. In general, I suggest carefully reading the manuscript and editing it to shorten the text.*

*Below, I list a few more comments that the authors may want to consider before publication. These are mostly suggestions and are up to the authors if they want to include them.*

We thank the Reviewer for their helpful comments and suggestions. We have removed approximately 600 words from the manuscript, primarily from sections 2–4, without altering the science content or message of our study. This includes significantly reducing the description of results in supplemental material (now on lines 457–467) that you've noted.

*Comments*

1. *129: 'radar-based proxy'*

   Modified to "…radiometer- and radar-based proxies…".

2. *416: 'is capturing'*

   Modified as suggested.

3. *Can the corresponding parts of Figure 2 shown in Figure 3 be indicated somehow? For example, by putting a thicker box in Figure 2 around the three values for Mean 355-nm Bsc binned by LCL shown in Figure 3a. The same can be done for Figures 4,6, and 7. That should make it easier to find the parts of the figures that correspond to each other.*

Boxes have been added to each of these figures, along with Figs. S1 and S3 in the supplemental material, to highlight which correlation sets are represented in the corresponding scatterplots, as suggested.

4. *In general, I would suggest flipping x-axis and y-axis in Figures 3,5, and 8 since technically the authors are investigating the convective variables as a function of the aerosol variables.*

We have flipped the x- and y-axes in each of these figures (along with Fig. S2 in the supplemental material) as suggested, and we have made minor updates to the figure captions to provide additional clarity about the scatterplots and data shown therein.  The x-axis range in Fig. 3a has also been updated to avoid erroneously cutting off data points with $Bsc_{355} > 6$ $Mm^{-1}$ $sr^{-1}$; all other scatterplots have been checked for this same mistake.  Also, in Fig. 1, we have increased font sizes of the y-axis labels for easier viewing and corrected the HSRL2 title from "355-nm" to "532-nm."

5. *525: K-index looks like it has the lowest correlations in Fig. 6.*

We have removed K-Index from this list (lines 503–504), which had been accidentally left in the list from a prior version of the manuscript.

**Reviewer #3**

*The authors clearly improved the manuscript through the rounds of the review process and eliminated major concerns about interpretations of the results (I agree that the results are inconclusive and do not present an indisputable trend). I have the following minor comments regarding the abstract and introduction sections:*

We thank the Reviewer for their helpful recommendations.

*- I object "directly related" in line 25 :"radar- and radiometer-based parameters directly related to convective intensity...". The authors also call them "indirect" indicators in lines 116 and 128, which I believe is more appropriate.*

We have modified this phrase to "radar- and radiometer-based parameters with physical implications for convective intensity…" on lines 25–26.

*- I believe line 74: "The impacts of aerosol warm-phase and cold-phase invigoration of convection have received mixed results in prior observational and numerical modeling studies." reads as if "aerosol warm-phase and cold-phase invigoration of convection" is a proven fact as was pointed out in previous review round. I suggest that this sentence is restated.*

We have modified this sentence (lines 74–75) to "Potential impacts from hypothesized aerosol warm-phase and cold-phase invigoration of convection have received mixed results in prior observational and numerical modeling studies.".

*- The following statements in the same paragraph (starting at line 74) describes previous observational or modeling studies. I think it is important and relevant information to state which work uses observations and which work uses modeling to arrive at the described result/conclusion.*

We have added a few words throughout this paragraph (lines 74–92) to note which studies primarily used numerical simulations and which studies focused on observations.